# Enabling Efficient Sizing of Hybrid Power Plants: A Surrogate-Based Approach to Energy Management System Modeling

Charbel Assaad[1], Juan Pablo Murcia Leon[1], Julian Quick[1], Tuhfe Göçmen[1], Sami Ghazouani[2], and Kaushik Das[1]

[1]Department of Wind and Energy Systems, Technical University of Denmark, Roskilde, Denmark
[2]TotalEnergies, Paris, La Défense, France

**Correspondence:** Charbel Assaad (chass@dtu.dk)

**Abstract.**

Optimal sizing of Hybrid Power Plants (HPPs), which include wind power plants and battery energy systems, is essential to prevent financial losses from under- or over-sizing relative to grid connection capacities. Accurate sizing requires high-fidelity Energy Management Systems (EMS) to model bidding strategies and operations in electricity markets, resulting in precise operational revenues and costs. However, due to the computational burden of such models, sizing methodologies often resort to low-fidelity EMS models, leading to faulty sizing evaluations. To address the need for a computationally efficient and accurate model that enables quantitative assessment of HPPs, we evaluate the potential of surrogate models to replace a high-fidelity EMS participating in the day-ahead electricity market in Denmark with perfect forecasts. Given the limited literature on surrogates of EMS models for utility-scale, grid-connected HPPs with batteries, we develop and compare four different surrogate models to approximate a state-of-the-art EMS model. The best-performing surrogate employs singular value decomposition for dimensionality reduction and a feed-forward neural network for regression. This surrogate achieves a normalized root mean square error of 0.81% in approximating yearly revenues. This method proves effective in accurately evaluating the operation of HPPs across various geographical locations and hence in multiple sizing problems. Furthermore, we utilize the surrogate to evaluate the profitability of several HPP sizes, achieving a root mean square error of 0.010 on the profitability index, with values ranging between -0.13 and 0.18. This demonstrates that the developed surrogate model is suitable for HPP sizing under the given cost and financial assumptions.

## 1 Introduction

As the renewable energy and storage industry matures, governmental incentives that sustain this rapid development have started to shift. Initially sustained by government-supported feed-in tariffs, wind power plants are transitioning to feed-in premiums or contracts-for-differences. As noted by Kitzing et al. (2024), contracts-for-differences have become increasingly significant in the European market, particularly following an agreement reached in May 2024 between the European Commission, the European Parliament, and the Council (Council of the European Union, 2024). This agreement mandates that the support

for renewable technologies is provided through two-sided contracts-for-differences or equivalent schemes, applying to both new investments and repowering. These support levels are now determined through competitive bidding procedures Busch et al. (2023). Additionally, the power plants are expected to maximize their value from energy markets such as the electricity spot market or balancing/reserve markets. This change exposes power plant developers to the dynamics of the wholesale electricity market. In this evolving landscape, Hybrid Power Plants (HPPs) that include storage technology are becoming valuable to establish robust business cases. Despite the absence of a universal definition for HPPs — highlighted by varying interpretations in the literature (Dykes et al., 2020; Long et al., 2022; Paska et al., 2009) — for the purposes of this research, we define renewable-based HPPs as power plants that combine several generation technologies, including wind turbines and possibly energy storage, to produce electricity and other energy vectors. They operate from a single geographical location, with all generated power being transmitted to the electrical grid via a unified point of grid connection.

In the presence of a BESS, HPPs can use strategies such as market arbitrage, which involves buying and storing electricity when market prices are low and selling it when prices are high. Additionally, the integration of energy storage is crucial for smoothing power supply fluctuations, mitigating power curtailment, enabling HPPs to offer system services, and reducing grid congestion (Das et al., 2019).

As HPPs transition to market-driven revenue models, optimizing their operations becomes essential to capitalize on market opportunities and address operational challenges. An EMS fulfills this role by devising optimal operational strategies throughout the power plant's lifetime. For an HPP with a BESS, the EMS optimizes the charge and discharge power of the battery, given the available resources (i.e., wind speed, solar power), the battery's capacity, available grid connection capacity, and other constraints. In electricity trading, various markets allow power plant operators to sell their energy. The Spot Market (SM) is currently the most lucrative market where power is traded for immediate delivery. Within the SM, operators can participate in the day-ahead and hour-ahead markets. Day-ahead bidding establishes hourly prices for the following day, while hour-ahead bidding allows for adjustments based on updated generation forecasts and cleared SM prices. Since electricity markets require power producers to bid in advance on the amount of electricity they will generate, these bids rely on forecasts of renewable energy generation and market prices. If actual generation deviates from scheduled bids, financial penalties are incurred. Additionally, the balancing market (BM) offers HPPs another revenue source, operating alongside the SM to handle discrepancies between forecasted and actual demand and supply. The BM allows transmission system operators to adjust for differences arising from SM bids that are based on earlier forecasts and the real-time conditions closer to delivery. This market acts in near real-time, addressing deviations from scheduled generation and imposing penalties as necessary. Consequently, an EMS aims to maximize profits by strategically storing and selling electricity in both the SM and BM, while also minimizing imbalance costs due to deviations from scheduled energy bids.

In the field of hybrid renewable energy systems, particularly in microgrids, the dispatch problem has been extensively studied, as evidenced by numerous review articles on the topic (Barbosa et al., 2024; Shivarama Krishna and Sathish Kumar, 2015; Fathima and Palanisamy, 2015). However, the primary purpose of microgrids is to manage or follow load profiles within a network, whereas HPPs operate as distinct generation facilities with a connection to the power grid. This unique connection

emphasizes the role of HPPs in active power generation, rather than load management alone, which calls for a distinct EMS model such as the ones developed by Toubeau et al. (2021) and Ding et al. (2016).

EMS models vary in complexity and computational demand, and for this article, we categorize them into high-fidelity (HF) and low-fidelity (LF) models. High-fidelity EMS models provide detailed and accurate representations of HPPs, capturing intricate system dynamics, precise component behaviors, and sophisticated market interactions. These models incorporate forecasting and real-time data, comprehensive operational constraints, and optimize bidding strategies to maximize profits in electricity markets (Taha et al., 2018; Zhu et al., 2024; Ochoa et al., 2022; Han and Hug, 2020; Li and Qiu, 2016; Abdeltawab and Mohamed, 2015; Yang et al., 2018; Huang et al., 2021; Das et al., 2020). While HF EMS models offer high accuracy in estimating operational performance and financial outcomes, they require significant computational resources and time. For instance, Huang et al. (2021) develops a stochastic optimization-based HF EMS where solving the dispatch for one week of operation takes between 329 and 2,991 seconds, depending on the chosen optimization algorithm among five compared. Similarly, Li and Qiu (2016) present a deterministic HF EMS model that requires using a monthly time step to reduce simulation costs.

In contrast, LF EMS models simplify the representation of HPP operations by using aggregated system models, basic forecasting methods (or none), and simplified market participation strategies (An et al., 2020; Luo et al., 2015; Cai et al., 2016; Zhang et al., 2018, 2017). They reduce computational demand by neglecting detailed component behaviors and operational constraints. Although they enable rapid simulations and are easier to integrate into optimization frameworks, their oversimplifications can lead to inaccuracies in revenue and cost estimations, potentially resulting in sub-optimal or erroneous sizing decisions. Table 1 provides an overview of the distinctions between HF and LF EMS models. Notably, there is no universally accepted standard for defining these classifications, so the table serves as a guideline based on key characteristics relevant to this study.

The trade-off between computational efficiency and model accuracy presents a significant challenge for the optimal sizing of HPPs. A sizing optimization of an HPP involves maximizing a financial metric by varying the wind power plant rating, battery energy, and power ratings. Calculating that financial metric requires solving an EMS model for each potential HPP configuration. Consequently, HF EMS models offer precise assessments; however, relying solely on them is impractical due to their substantial computational demands. Conversely, using LF EMS models reduces computational time but risks compromising the financial viability of the project due to inaccurate assessments. To illustrate the computational burden of an HF EMS model, we evaluate the state-of-the-art EMS developed by Zhu et al. (2022). This model requires 1,250 minutes to solve for 25 years of operation (the assumed lifetime) of a given HPP using a single-node High Performance Computing (HPC) cluster, Sophia (DTU HPC Cluster, 2019), which has 32 physical cores (2 × sixteen-core AMD EPYC 7351) and 128 GB of RAM (4 GB per core, DDR4@2666 MHz). Therefore, even if we need to evaluate only a few sizings for the optimizer to converge, we require a substantial amount of time to reach a solution. For example, evaluating 10 sizings takes 12,500 minutes, or approximately 208 hours. Additionally, in previous work familiar to the authors (Leon et al., 2024), a sizing optimization can take up to several hundred iterations to approach optimality. In that study, the authors use a low-fidelity EMS model to evaluate the operation of an HPP over its lifetime in a matter of 15 seconds. The comparison of the optimization time is based

**Table 1.** Comparison of HF and LF EMS Models

| Aspect | HF EMS | LF EMS |
|---|---|---|
| Component modeling | Physical modeling per component considering the electrical, mechanical, and/or thermal behaviors. Includes battery degradation model | Simplified or aggregated model, linear approximations or average values used for component performance |
| Market Participation Modeling | Complex bidding strategies in various electricity markets (day-ahead, intraday, balancing), includes market rules and regulations (i.e., dispatch and settlement intervals), and/or grid compliance requirements | Usually focuses on one market, ignoring opportunities or penalties from other markets |
| Input data and Forecast | Varying resolution (from sub-hourly to hourly). Possible combination of forecast and real-time data | Uses hourly resolution. Deterministic forecasts or perfect forecast |
| Optimization problem formulation | Complex formulations like: mixed-integer linear programming (MILP), nonlinear programming (NLP), or stochastic optimization | None e.g. rule-based, or linear programming |
| Computational Demand | High computational demand due to complex modeling and optimization, leading to long simulation times | Low computational demand, allowing for fast simulations and scalability |
| Accuracy | High | Low |

on the same computational resources. Given these computational benefits, HPP sizing optimization often relies on LF EMS models. For example, Leon et al. (2024) propose a methodology for sizing HPPs as a nested optimization problem, using two LF EMS models: a short-term EMS formulated as linear programming and a long-term rule-based EMS. The short-term EMS provides a baseline for daily optimal operations, while the long-term EMS modifies these operations to account for degradation effects and forecast inaccuracies over the plant's lifetime. Similarly, in a study aimed at optimizing the design and layout of a hybrid wind-solar-storage plant, Stanley and King (2022) employs a simple battery dispatch model, where the battery is only discharged to meet minimum power requirements. While using LF EMS models may result in reduced accuracy in revenue estimation, they are widely adopted in HPP sizing due to computational efficiency. Indeed, several review studies underscore the prevalence of LF EMS models in sizing methodologies (Roy et al., 2022; Lian et al., 2019; Thirunavukkarasu et al., 2023).

It is challenging to quantify the accuracy loss when using LF EMS instead of HF models. Research studies often test EMS models on varied configurations of HPPs, and only a few conduct direct comparative analyses within the same setup, primarily focusing on high-fidelity EMS models. For instance, Ochoa et al. (2022) compare deep reinforcement learning with both stochastic optimization and robust optimization for photovoltaic-battery HPPs using U.S. market data, finding that deep

reinforcement learning offers superior economic performance and significantly reduces computational time compared to the other two studied methods. Similarly, Han and Hug (2020) report that, in a one-year simulation using Nord Pool data, the distributionally robust optimization model achieves higher revenues than deterministic forecasting approaches. Zhu et al. (2024) further explore this by comparing EMS models that utilize distributionally robust optimization with those based on deterministic optimization and stochastic optimization for wind-battery hybrid plants in Nordic day-ahead markets, taking imbalance settlements into account. By adjusting the parameters of the distributionally robust optimization model, they demonstrate that the economic performance ranks highest for this approach, followed by risk-neutral stochastic optimization, and finally deterministic optimization. This approach enables more resilient offering strategies, especially in markets with high penalties for energy imbalances. Additionally, Zhu et al. (2024) examine the accuracy of total profits across three HF EMS models and show that even within these models, certain simplifications commonly found in low-fidelity models—such as the use of deterministic forecasts—can lead to revenue discrepancies of up to 7.6% compared to the best-performing model (refer to Table 3 of the referenced paper). Given these considerations, this paper primarily focuses on reducing the computational demand of high-fidelity EMS models.

To address the computational challenges associated with implementing a realistic EMS for HPP sizing while maintaining high accuracy, a promising approach involves using data-driven surrogate-based modeling. This technique demonstrates potential in tackling computationally intensive problems across various domains (Zhang et al., 2021; Lin et al., 2023; Pang et al., 2023). These Reduced-Order Models (ROM) aim to replace high-dimensional, resource-intensive problems with models that are significantly faster to simulate while accurately representing the original solution behavior. In particular, Hesthaven et al. (2022) reviews the development of surrogates for time-dependent problems, including those with nonlinear dynamics, which are of interest in our work. In this context, data-driven surrogate models stand out as promising solutions, thanks to major advancements in machine learning methods. These models often follow an offline-online paradigm. During the offline phase, a reduced basis is extracted from a collection of high-fidelity solutions; this reduced basis is then used to train the surrogate model by optimizing weights or coefficients that capture the system dynamics. Although this step is computationally intensive, it only needs to be performed once. In the online phase, the surrogate model uses the precomputed weights to compute new outputs almost instantly, with minimal computational cost. This paradigm enables the surrogate model not only to learn the mapping from inputs to outputs but also to understand underlying patterns within the input data, leading to faster and more accurate simulations. Numerous successful implementations of data-driven surrogate models exist in the literature. For instance, Hesthaven and Ubbiali (2018) develops an ROM using Proper Orthogonal Decomposition (POD) to extract a reduced basis from high-fidelity solutions and employs multi-layer perceptron neural networks to approximate the coefficients of the reduced model, although time-dependency is not considered. Similarly, Guo and Hesthaven (2019) uses a POD projection and maps the time and parameter values onto the reduced basis using tensor products of two Gaussian processes—one for time and one for parameters. Hess et al. (2023) utilizes a data-driven ROM approach to efficiently compute the Rayleigh–Bénard cavity problem, integrating POD, dynamic mode decomposition, and manifold interpolation for a robust and computationally efficient model. Departing from POD, Bhatt et al. (2023) employs deep auto-encoder networks to compress high-fidelity snapshots before using these in forecasting models—specifically, long short-term memory and temporal convolutional networks for

time-series forecasts, and convolutional neural networks for spatial feature extraction—significantly reducing computational costs in both the offline and online stages. Most ROMs have been applied to problems described by partial differential equations with sharp gradients. In contrast, our aim is to apply similar techniques to high-fidelity EMS models for HPPs. Although surrogate modeling has advanced across multiple fields, a gap remains in developing models tailored to EMS for utility-scale HPPs, particularly those that incorporate detailed operational strategies for market participation. This gap exists not only due to the scarcity of existing applications of surrogate models for EMS in HPPs but also because of the complexity involved in designing an accurate surrogate model based on a multitude of input and output time series. Additionally, integrating the surrogate model within a sizing evaluation framework adds another layer of complexity.

This article seeks to evaluate the potential of data-driven surrogate models in reducing the computational burden of high-fidelity EMS models, while preserving the high accuracy needed for a reliable assessment of HPPs. To achieve this, we develop four surrogate models to approximate the outputs of the HF EMS. We begin by employing two models based on multivariate linear regression, which serve as baselines due to their simplicity, interpretability, and low computational cost. Recognizing that the EMS exhibits complex and nonlinear behaviors that linear models may not capture adequately, we also develop two models based on Neural Networks (NNs). Neural networks, particularly Feed-forward Neural Networks (FNNs), are capable of modeling intricate nonlinear relationships through their layered architectures and nonlinear activation functions. This makes them well-suited for approximating the inherent nonlinear dynamics of the HF EMS. By leveraging NNs, we can capture complex patterns and interactions within the data that linear models might overlook, potentially achieving higher accuracy in estimating outputs. The choice of these surrogate models allows us to explore the trade-offs between model complexity, computational efficiency, and accuracy. By comparing the performance of multivariate linear regression models and neural networks, we assess the extent to which incorporating nonlinearity improves the surrogate's ability to replicate the HF EMS outputs. Additionally, NNs have demonstrated success in surrogate modeling across various fields due to their flexibility and scalability, making them a promising candidate for this application.

Building upon the EMS model developed by Zhu et al. (2022), which is detailed in the following section, this paper seeks to answer the question: How can we enable the sizing evaluation of utility-scale HPPs based on an accurate and computationally efficient EMS model?

The major contributions of this article are as follows:

– Development of a fast and accurate surrogate model for a utility-scale HPP EMS participating in the spot market.

– Demonstration of the surrogate's generalizability in different geographical locations within the same electricity market region.

– Integration of the developed surrogate within a sizing evaluation framework to accurately assess the profitability of various HPP configurations.

The remainder of this paper is organized as follows: Section 2 the HF EMS model that the surrogates are based on and the methodology devised for the surrogate modeling of the EMS. Section 3 details the sizing evaluation framework for analyzing

the profitability of an HPP using the HF EMS and the surrogate models. Section 4 provides details on the case study, the data used to train and validate the surrogate models, while Section 5 offers an in-depth analysis of the best-performing surrogate and its application. The performance of all surrogates is detailed in Appendix A. These results are put into perspective in Section 6 and summarized in Section 7.

## 2 Methodology

In this section, we begin by presenting the high-fidelity model that serves as the foundation for the surrogate models in Section 2.1, followed by the methodology for developing the four surrogate models in Section 2.2.

### 2.1 HPP Operation Model (EMS)

The EMS model, on which the surrogate is built, is presented in this section. The EMS model is based on a study by Zhu et al. (2022) that focuses on a co-located wind-battery HPP. This novel EMS model is formulated to optimize market participation within two sequential electricity markets: the SM and the BM, which encompass the regulatory periods of the Danish market structure. This state-of-the-art EMS has the advantage of considering

– Long-term operation of the HPP with comprehensive revenue stream calculations.
– Grid capacity as a practical constraint for the HPP.
– The possibility of considering overplanting, which has been shown to increase the value of HPPs.

However, this paper primarily focuses on the EMS's role in day-ahead SM participation. Additionally, our study considers the Danish market structure with a dispatch interval of 15 minutes. As in real power plants, the SM bidding process (also referred to as SM optimization) communicates with a Real-Time (RT) dispatch optimization. In this framework, the SM optimization provides energy set-points based on weather and market forecast data to the RT dispatch, which, in turn, uses real-time measurement data to derive real-time power values. Real-time measurements allow the calculation of deviations and the application of penalties. The inputs to the SM optimization are time series forecasts of wind power and market prices, while the RT dispatch uses the same input time series updated with real-time measurements for each dispatch interval, as well as the bidding schedule generated from the SM optimization. For clarity, the inputs and outputs of the SM optimization and RT dispatch are listed in Table 2. The combined models, SM optimization and RT dispatch, are referred to as a high-fidelity EMS model in this paper.

While the SM optimization's input and output time series are based on hourly time steps, the RT dispatch's outputs and real-time input time series have a time step equal to the dispatch interval, i.e., 15 minutes for the Danish market structure. Additionally, both models assume a given HPP configuration, also referred to as sizing parameters in this article. These are defined as the wind power plant rated power ($P^W$), the rated battery power ($B^P$), battery energy capacity ($B^E$), and grid connection power capacity ($P^G$). Other battery parameters, such as charge/discharge efficiency, are assumed from Zhu et al. (2022).

**Table 2.** Inputs and Outputs of SM optimization and RT dispatch from Zhu et al. (2022).

| Model | Inputs | Outputs |
|---|---|---|
| SM optimization | HPP configuration<br>Wind power forecast<br>SM price forecast | HPP power output schedule<br>Battery charge/discharge power<br>Battery state of charge |
| RT dispatch | Same as SM optimization, and<br>RT wind power<br>Cleared SM prices<br>SM optimization power output schedule | RT HPP power output<br>RT battery charge/discharge power<br>RT battery state of charge<br>RT curtailed power |

Figure 1 illustrates the considered EMS workflow. The EMS operates through a structured daily cycle, beginning with the forecasting stage. On the previous day (d-1), a forecast of wind generation and spot market prices for the following day (d) is obtained. Using this data, the SM optimization is conducted at noon on day d-1, aligning with the day-ahead market closure, to determine the optimal hourly power bidding for the HPP. This optimization is formulated as a Mixed Integer Linear Programming (MILP) problem, aiming to maximize the plant's revenue across the day by strategically bidding power on the SM. On day d, the RT dispatch optimization is executed at 15-minute intervals throughout the day. This phase focuses on minimizing discrepancies between the power that was bid on the spot market and the actual real-time available power. The RT dispatch is modeled as a Mixed Integer Quadratic Programming (MIQP) problem, which dynamically adjusts the HPP's output to meet market commitments as closely as possible, responding to variations in generation. Finally, the day concludes with a settlement process on day d+1, where the outcomes of the day's operations are reconciled. The details of all models can be found in the referenced work.

The optimization models were solved using the solver of IBM Decision Optimization Studio CPLEX through the docplex Python library (IBM, 2023), operating on DTU's high-performance computing cluster, Sophia (DTU HPC Cluster, 2019). It was observed that for a given HPP configuration, 47 minutes were required to compute the outputs for one year of operation of the HF EMS model. The underlying reason for this is due to the iterative and sequential nature of the framework. For each day, the MILP optimization is solved first, followed by the MIQP for each dispatch interval (e.g., 96 times per day). While each iteration of the MILP and MIQP problems requires a minimal amount of time (less than 0.15 seconds), the frequency of these optimizations is substantial. Moreover, since each time step depends on the previous one, it is necessary to perform the optimization sequentially. Table 3 shows the number of decision variables and constraints required to optimize for inputs spanning over one year. This highlights the substantial computational time required to optimize the sizing of an HPP based on such an operational model.

While the combination of both models allows for a realistic representation of the operation of an HPP, it has its own limitations. These limitations are also carried over to the surrogate that is built upon both models. No battery degradation model is considered in the optimization process. It is well known that lithium-ion batteries' energy capacity degrades over time in a

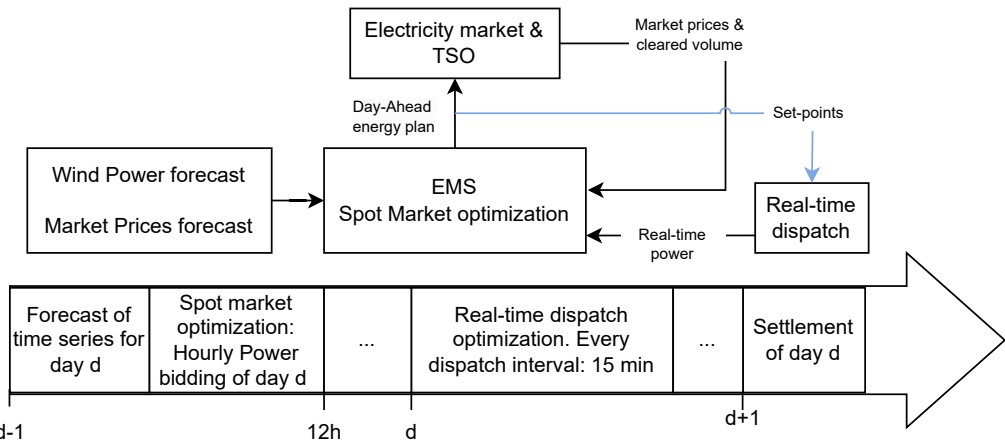

**Figure 1.** Workflow of HF EMS of this article, developed by Zhu et al. (2022)

**Table 3.** Computational Complexity of HF EMS' models for one year of input data

| Model | Design variables | Constraints |
|---|---|---|
| SM optimization | 289,445 | 350,765 |
| RT dispatch | 315,360 | 420,480 |

nonlinear fashion (Xu et al., 2018), which directly impacts revenues as opportunities for energy arbitrage are reduced. Additionally, as we focus on demonstrating the potential of surrogate modeling for EMS of HPPs, perfect forecast data will be used. Finally, the balancing market will not be considered in this article and will be left for future work.

## 2.2 Surrogate Methodology

In this article, a surrogate model consists of several sub-components: data pre-processing, a regressor, and data post-processing. The pre-processing involves scaling, which ensures that all inputs contribute equally to the model's estimations and supports the surrogate's convergence algorithm. The post-processing is applied in accordance with the pre-processing to interpret the results in their original scale. The regressor is the model tasked with approximating the high-fidelity EMS. Section 2.2.1 details the inputs and outputs for training and evaluating the surrogate models. Section 2.2.2 describes four surrogate models, differing in their data processing and regressor models. Sections 2.2.3 and 2.2.4 cover the training and validation of these models, respectively.

### 2.2.1 Surrogate Model's Inputs and Outputs

Regardless of the surrogate model being evaluated, all models aim to approximate the same output time series given the same input data. Table 4 below lists the various input and output time series used to train and validate the surrogate.

**Table 4.** Input and output time series of surrogate models

|  | Variable | Time step | Time horizon |
|---|---|---|---|
| Input | SM price forecast: $SM_t$<br>Wind power forecast: $W_t$ | 15 min | 1 day |
| Output | HPP power output: $P_t^{HPP}$<br>Battery discharging power: $P_t^{dis}$<br>Battery charging power: $P_t^{cha}$<br>HPP curtailed power: $P_t^{curt}$ | 1 hour | 1 day |

The input time series of the surrogate match those of the SM optimization, and its outputs align with the RT dispatch's outputs. In addition to the input time series, the regressor also considers three scalar parameters representing an HPP, which help differentiate between various HPP configurations. These parameters are represented by the ratios $P^W/P^G$, $B^P/P^G$, and $B^E/B^P$.

### 2.2.2 Surrogate Models

Table 5 presents all surrogate models evaluated. The first surrogate, S1, which serves as our benchmark, normalizes the input and output time series for each HPP configuration and employs multivariate linear regression to compute the normalized outputs from normalized inputs. The second surrogate, S2, incorporates a dimensionality reduction method known as Singular Value Decomposition (SVD), as developed by Gene H. Golub (1996). After normalizing inputs and outputs, we apply SVD, a common tool in numerical analysis, particularly for dimensionality reduction. The specific use of this method is detailed in this section. Surrogate models S3 and S4 are similar to S1 and S2, but they differ in their choice of regressors. Instead of employing a multivariate linear model, these models utilize a tuned FNN to capture the non-linear relationships between inputs and outputs.

For all surrogate models, we apply data post-processing consistent with the pre-processing to ensure the output data is interpretable in its original scale. For all surrogate models, we normalize the wind power generation time series ($W_t$) using the turbine's rated power and the spot market price time series ($SM_t$) by the maximum price, achieving a scaling between zero and one. Since the output time series magnitudes depend on the sizing inputs, we use these parameters as the basis for

**Table 5.** Surrogate models tested

| Model Name | Pre-processing | Regressor | Post-processing |
|---|---|---|---|
| S1 | Normalization | Linear | Reverse Normalization |
| S2 | Normalization SVD | Linear | Reverse SVD Reverse Normalization |
| S3 | Normalization | FNN | Reverse Normalization |
| S4 | Normalization SVD | FNN | Reverse SVD Reverse Normalization |

normalization following these equations:

$$P_{t,norm}^{HPP} = P_t^{HPP}/P^G \tag{1}$$

$$P_{t,norm}^{dis} = P_t^{dis}/B^P \tag{2}$$

$$P_{t,norm}^{cha} = P_t^{cha}/B^P \tag{3}$$

$$P_{t,norm}^{curt} = P_t^{curt}/(P^W - P^G) \tag{4}$$

We apply the SVD as described in Zhu et al. (2010) to derive the principal component matrices. This method is used independently for the matrices containing the input time series $M_{in}$ and the output time series $M_{out}$. The SVD is applied following the normalization described above and is used for surrogate models S2 and S4. Figure 2 illustrates the matrix $M_{in}$, which includes all normalized input time series for a single HPP configuration: the normalized wind power time series for day $d$ and for the HPP configuration $n$, $W_{t,norm,d}^{HPP_n}$, and the normalized SM prices $SM_{t,norm,d}^{HPP_n}$.

One HPP configuration = 365 Days

$$M_{\text{in}} = \begin{pmatrix} W_{t,norm,1}^{HPP_1} & W_{t,norm,2}^{HPP_1} & \cdots & W_{t,norm,365}^{HPP_1} \\ SM_{t,norm,1}^{HPP_1} & SM_{t,norm,2}^{HPP_1} & \cdots & SM_{t,norm,365}^{HPP_1} \end{pmatrix}$$

Time steps in one day

**Figure 2.** Matrix M containing input time series, denoted $M_{in}$. $W_{t,norm,d}^{HPP_n}$ is the normalized wind power time series for day $d$ and for the HPP configuration $n$. $SM_{t,norm,d}^{HPP_n}$ is the normalized SM prices

As the high-fidelity EMS uses hourly time steps for forecasted wind power and SM prices, each of the input vectors, $SM_{t,norm,d}^{HPP_n}$ and $W_{t,norm,d}^{HPP_n}$, has 24 time steps. To expand this matrix for all HPP configurations, we concatenate horizontally (i.e., along the second dimension) each matrix $M_{in}$ corresponding to an HPP configuration. The output time series matrix,

$M_{out}$, is constructed in a similar fashion. However, unlike $M_{in}$, this matrix contains four time series, the ones defined in Eq. (1) to (4). Note that these time series have a time step equal to the dispatch interval, e.g., 15 minutes.

After applying Singular Value Decomposition (SVD) to both matrices, $M_{in}$ and $M_{out}$, we extract their principal component matrices and truncate them to the desired level. As a result, we obtain two sets of matrices with different truncation levels, denoted as $r_{in}$ and $r_{out}$, respectively. The truncation level is chosen so that the explained variance is 99%; for the definition, see Eq. 4 of Freire and Ulrych (1988). Table 6 presents an overview of the features and samples of each data-processing method for input and output data spanning over a year.

**Table 6.** Features and Samples of Data-processing methods

| Data processing | | Normalization | SVD |
|---|---|---|---|
| Inputs | Features | 5 | $r_{in} + 3$ |
| | Samples | $(24 \cdot 2) \cdot 365 \cdot N$ | $365 \cdot N$ |
| Outputs | Features | 4 | $r_{out}$ |
| | Samples | $(96 \cdot 4) \cdot 365 \cdot N$ | $365 \cdot N$ |

From this table, we observe that the features of the SVD method are likely higher than those derived from the normalization method (this depends on the truncation level). However, the number of samples is substantially lower, which allows us to achieve a reduced representation of the data. Note that the surrogate models using only normalization for data processing (S1 and S3) have five input features:: $SM_t$, $W_t$, $P^W/P^G$, $B^P/P^G$, and $B^E/B^P$. And they output four features: $P_t^{HPP}$, $P_t^{dis}$, $P_t^{cha}$, and $P_t^{curt}$.

### 2.2.3 Surrogate Training

To train a surrogate model, a training dataset is defined based on a number of HPP configurations with distinct sizing parameters. The details of this dataset can be found in Section 4. More specifically, this training dataset is used to train the SVD transformation and the two regressor models. Note that normalization does not require training. Two models are used to approximate the outputs of the high-fidelity EMS: a tuned FNN (models S3 and S4) and a multivariate linear regression (models S1 and S2). The latter is used as a baseline model to compare the accuracy of the neural networks.

Since the regressors used in S1 and S2 differ significantly from those in S3 and S4, their training processes also vary. Models S3 and S4 use an FNN; hence, the training is done in two steps and applied for each of models S3 and S4 individually. The first step involves a tuning process and is carried out using two hyperparameters, shown in Table 7 below. Within this tuning process, it's important to note that each layer can have a varying number of neurons within the provided range. Afterwards, the best-performing model from the tuning process, for each of S3 and S4, is selected for more exhaustive training. This two-step approach is necessary to reduce the computational burden introduced by the tuning process, in which several hundred NN architectures are evaluated; however, the NNs aren't trained until they converge. Instead, the best-performing model from the

tuning process is selected for further training. To efficiently select the hyperparameters among the search space, Hyperband by Li et al. (2018) is used. Hyperband uses random sampling of hyperparameters to explore a wide range of settings. For both models S3 and S4, a Rectified Linear Unit (ReLU) activation function is used for all hidden layers, and for the output layer, a linear activation function is used. ReLU is an appropriate activation function for the data, particularly following the normalization process, as all input and output time series become non-negative. To train an NN, at least two settings need to be defined. First, a loss function, which measures the error between the training data and the model's estimations—the mean squared error is used. Second, an optimizer, which modifies the model's weights and biases during the training process to minimize the loss function. Each optimizer has its own set of hyperparameters. The Adam optimizer by Kingma and Ba (2017) is used with a learning rate of $10^{-4}$. The tuning results in the architectures presented in Table B1 and B2 in AppendixB.

**Table 7.** FNN grid search hyperparameter space

| Hyperparameter | Range | Step |
|---|---|---|
| Layers | [3,9] | 1 |
| Neurons per layer | [40,80] | 20 |

Models S1 and S2 use multivariate linear regression. They are trained using the same dataset with the objective of minimizing the mean squared error, using the same optimizer as for the FNN. Models S1 and S2 are not subject to any tuning, as there are no hidden layers or neurons. Similarly, the SVD transformations are trained on the training dataset. The transformations are trained separately for both input and output time series, resulting in two distinct transformations.

### 2.2.4 Surrogate Validation

We aim to identify the surrogate model that offers the best compromise between training time, inference time, and accuracy. First, we assess each surrogate's accuracy on a validation dataset, which is separate from but defined similarly to the training dataset. We measure the accuracy of each surrogate using the Root Mean Square Error (RMSE) between the estimated ($\hat{y}_{i,norm}$) and actual ($y_{i,,norm}$) values for the normalized hourly time series and for all data points in the validation dataset ($N$). The RMSE ($\epsilon_{RMS}$) is computed as follows:

$$\epsilon_{RMS} = \sqrt{\frac{1}{N}\sum_{i=1}^{N}(y_{i,norm} - \hat{y}_{i,norm})^2} \qquad (5)$$

Since this RMSE is calculated across all output time series, it provides a broad assessment of the model's accuracy, without specific insights into each individual series. We use this metric to compare the performance of the surrogate models presented in Table 5. Additionally, we measure the training and inference time of each model.

For the best-performing surrogate model, among S1-S4, we further investigate the accuracy of each output time series using RMSE for a deeper understanding of the model. Our focus then shifts to one specific output time series, the normalized power

output $P_{t,norm}^{HPP}$. This time series allows us to calculate the yearly revenues, which are required to compute the Profitability Index (PI), enabling us to evaluate the profitability of a given sizing, the key application of our surrogate in this article.

To explore the methodology's potential further, we assess the surrogate's generalizability across various locations within the western Danish price region, DK1. In this intra-generalizability analysis, we calculate the Normalized Root Mean Square Error (NRMSE) of the yearly revenues to assess the accuracy of the surrogate's approximations. Specifically, we compare the approximated revenues ($\hat{\Pi}_k$) with the actual revenues from the HF EMS ($\Pi_k$) for each HPP configuration $k$ across all configurations in the selected set $K$. To express the RMSE as a relative measure, we normalize it by dividing by the median of the yearly revenues for all selected HPP configurations ($Median(\Pi)$), as computed by the HF EMS. The NRMSE ($\epsilon_{NRMS}$) is given as:

$$\epsilon_{NRMS} = \frac{\sqrt{\frac{1}{K}\sum_{k=1}^{K}(\Pi_k - \hat{\Pi}_k)^2}}{Median(\Pi)} \tag{6}$$

The revenue time series is extracted from either true/observed data (from the high-fidelity model) or approximated (from the surrogate model) power time series, considering the dispatch interval $\Delta t$ and the total time steps within a year, $T$.

$$\Pi = \sum_{i=t}^{T} P_t^{HPP} \cdot SM_t \cdot \Delta t \tag{7}$$

## 3  Application: PI Evaluation

To assess the business case of an HPP, we can use financial metrics like Internal Rate of Return (IRR) and Net Present Value (NPV). IRR calculates the HPP's annual investment return, while NPV assesses its profitability in today's value. However, when an HPP isn't profitable, resulting in a negative NPV, the IRR becomes undetermined. A more meaningful measure is the Profitability Index (PI), calculated as $NPV/CAPEX$. The PI indicates how many dollars of present value benefit are generated per dollar of investment, offering a more intuitive understanding of the investment's profitability. This metric allows for a direct comparison of the relative profitability of each project, regardless of their absolute size. Additionally, when resources are limited, $NPV/CAPEX$ can aid in prioritizing projects. Projects with higher PIs can be prioritized as they promise greater returns per unit of investment. A PI greater than 1 signifies that the NPV of future cash flows exceeds the initial investment. Note that traditionally, for power plants using one type of generation technology, the Levelized Cost of Energy (LCoE) is used to evaluate profitability. However, to assess the various potential revenue streams, metrics such as NPV (Dykes et al., 2020) or the NPV/CAPEX are more relevant. This is because storage inherently increases costs and thus the LCoE, even though it has the potential to substantially increase revenue or profit.

To compute the PI, we require the NPV, which in turn requires accurate yearly revenues and costs over the HPP's lifetime, aligning with the ideal framework shown in Fig. 3(a). Yet, as discussed in the Introduction and Section 2.1, this method is computationally demanding. We instead use an alternative framework in Fig. 3(b), utilizing the developed surrogate. This surrogate replaces the high-fidelity EMS, significantly reducing computational time and making the framework's execution feasible. The accuracy of this framework, employing the surrogate model to evaluate the PI, is presented in Section 5.4.

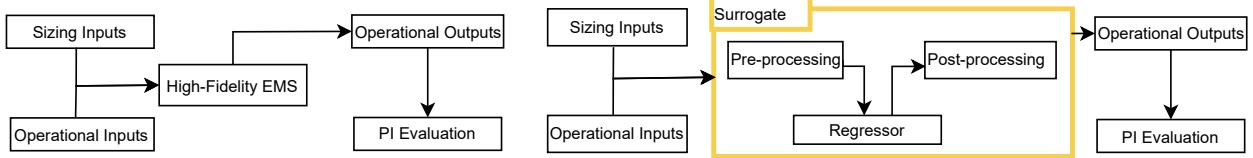

**Figure 3.** High-level sizing framework. Left: Ideal sizing evaluation framework using the High-Fidelity EMS. Right: Developed sizing evaluation framework using a surrogate of the EMS.

The PI varies between each sizing configuration (denoted as $x$), represented by $P^G, P^W, B^E, B^P$. PI is calculated as follows:

$$PI = \frac{NPV(x)}{CAPEX(x)} \tag{8}$$

The financial model is based on the yearly cash flow ($CF_y$) and the discount rate after tax ($r_{AT}$), throughout the lifetime of the power plant $Y$.

$$NPV = \sum_{y}^{Y} \frac{CF_y}{(1 + r_{AT})^y}$$

Where $Y$ is the lifetime of the power plant. The cash flow is calculated based on yearly profits ($P_y$) and CAPEX.

$$CF_y = \begin{cases} P_y & \text{for } y > 0 \\ -CAPEX & \text{for } y = 0 \end{cases}$$

It is important to highlight that the yearly profits are based on the revenues from the surrogate or the high-fidelity EMS ($\Pi_y$), as well as, the Operational Expenditure ($OPEX_y$), the tax rate ($\tau_{tax}$), and the $r_{AT}$.

$$P_y = (\Pi_y - OPEX_y) \cdot (1 - \tau_{tax})$$

The cost model used to calculated the $CAPEX$ and $OPEX$ is described in Appendix C. It should be noted that the calculation of NPV/CAPEX requires only the HPP power output time series from either the high-fidelity model or the surrogate.

## 4   Case Study

In Section 4.1, we introduce the training and validation dataset. Following this, Section 4.2 discusses the data related to the intra-generalizability analysis. In Appendix C, the cost model is presented, and the related data is provided in Table D2 of Appendix D.

## 4.1  Training and Validation Dataset

### 4.1.1  HPP configurations

As we rely on a surrogate to replace the high-fidelity EMS, we require a comprehensive dataset to train and validate this surrogate. Therefore, a wide range of HPP configurations needs to be covered. In addition, these configurations must be realistic and in line with industry practices. Table 8 summarizes the parameter ranges. For this article, the grid connection varies between 50 MW and 700 MW.

**Table 8.** Sizing Parameters and Ranges

| Sizing Parameter | Range |
| --- | --- |
| $P^W/P^G$ [-] | [1, 2] |
| $B^P/P^G$ [-] | [0, 1] |
| $B^E/B^P$ [h] | [1, 8] |

To ensure an equal distribution of all variables across the entire parameter space, the Latin Hypercube Sampling method, by Jin et al. (2005), is used to randomly select 250 sizing configurations, of which 200 HPP (80%) are used to train the regressor and 50 HPP (20%) are used to evaluate the accuracy of the surrogate, as detailed in Section 2.2.2. Subsequently, the high-fidelity EMS is solved using these configurations with the input time series presented in the section below.

### 4.1.2  Input time series & WT technology

The input time series required for the high-fidelity model, mentioned in Table 2, are generated using two tools. Wind power time series are simulated with the CorRES simulation tool (Murcia Leon et al., 2021; Koivisto et al., 2019). This tool is based on meteorological reanalysis data from the Weather Research and Forecasting model. CorRES' stochastic model (Koivisto et al., 2020b) is integrated to add fluctuations, resulting in wind power time series with 15-minute resolution. The simulation is based on meteorological data from the year 2012, with the assumption that the climate in 2030 remains unchanged from 2012. CorRES requires specific inputs, including the HPP's longitude, latitude, hub height of the wind turbine, power curve, and the simulation period. The considered turbine is the Gamesa G80 with a rated power of 2MW and a hub height of 100 meters.

SM price time series for the 2030 electricity markets are obtained from the Balancing Tool Chain (BTC) (Kanellas et al., 2020). BTC is built upon Balmorel, an open-source energy system model (Wiese et al., 2018) that simulates electricity market operations, ranging from day-ahead to real-time dynamics for the northern central European region. Additionally, an investment optimization is implemented to simulate a 2030 energy system scenario (Koivisto et al., 2020a).

### 4.1.3 Output time series

For all 250 HPP configurations and the aforementioned input time series, the high-fidelity model is used to generate all output time series described in Table 4. Out of these HPP configurations, 200 are used for training the surrogate model, and the remaining 50 are used to validate the model.

### 4.2 Intra-generalizability analysis Data

For the intra-generalizability analysis, we use the best surrogate model following the methodology described in Section 2.2. We then test the surrogate's accuracy across four randomly chosen locations within the same market region, labeled A to D. At each location, we randomly select 10 HPP configurations from the training dataset and another 10 from the validation dataset. The coordinates of each location are listed in Table D1. Figure 4 displays these locations, indicating the training location with an "X" and the evaluation sites for the HPPs. As all locations are in the same market region, the SM price time series is the same for all locations. The weather data for locations A-D is provided by CorRES, and the output time series per location and per HPP configuration are generated using the high-fidelity EMS model. This data is used to compare the performance of the surrogate trained on location X and evaluated on locations A-D.

The wind generation distribution across all locations is available in Fig. 4. This violin plot illustrates the distribution of normalized wind power generation across five different locations (X, A, B, C, and D). Each half-violin represents the density of wind power measurements for a location, showing where values are most concentrated. The symmetrical nature of each violin plot, with mirrored halves for each location, is a standard feature of violin plots that allows for a clearer visualization of the data distribution, where each half represents the same distribution of wind power measurements. The width of each violin indicates the density: wider sections reflect more frequent occurrences of those power levels, while narrower sections suggest less common values. The plot uses a logarithmic scale on the y-axis, making it possible to visualize variations in power generation across a broad range, from very low to high outputs. Inside each violin, the black bar marks the interquartile range, while the white dot represents the median of the wind power measurements for that location. This combination allows for a clear comparison of both the range and central tendencies of wind power output across different sites. For example, a location with a narrower and higher median distribution might experience more consistent and higher wind power generation (i.e., location X), while one with a broader distribution and lower median could have more variability (i.e., location C). Locations A, B, and D share similar distributions where the shape of these distributions suggests that low power output is more common, with occasional rises to higher values.

## 5 Surrogate Results

This section details the accuracy of the best surrogate model and its main application in evaluating the PI of HPPs. The comparison of all surrogate models is provided in Appendix A. After introducing the results of the most accurate surrogate model in Section 5.1, we examine how the accuracy of that model varies with different training dataset sizes in Section 5.2.

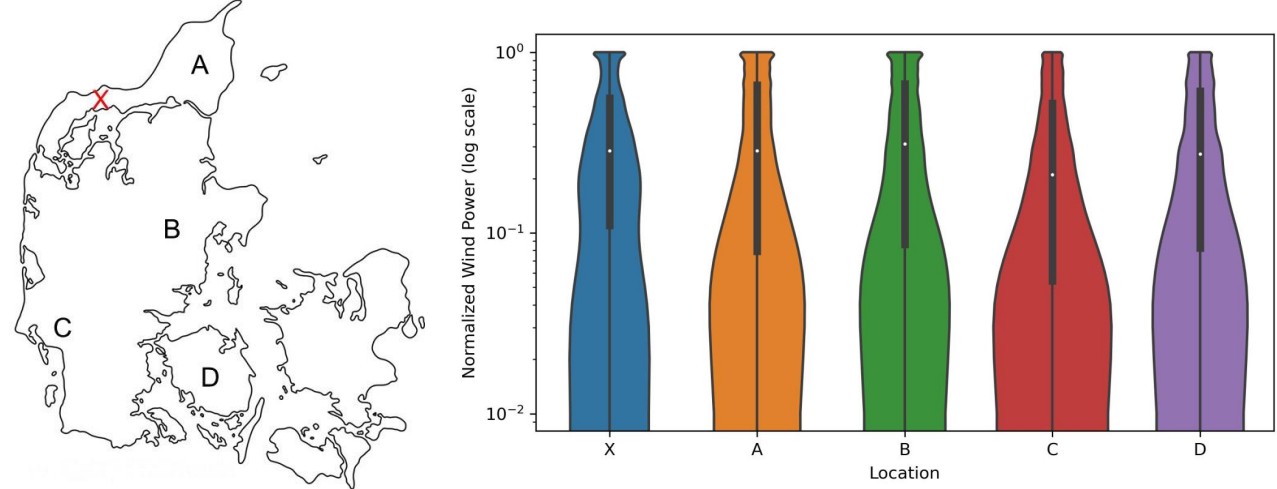

**Figure 4.** Intra-generalizability data. Left: location of trained surrogate model, X (in red) and evaluated model (A-D). Right: normalized wind power distribution across all locations. A log scale was used to highlight the difference between locations.

Next, in Section 5.3, we assess the surrogate's performance across various locations where it hasn't been trained. Finally, Section 5.4 compares the PI accuracy when evaluated using both the surrogate and the high-fidelity EMS. For all the results shown in this section, the validation dataset was used to evaluate the accuracy of the surrogate model and its application.

### 5.1 Best Performing Surrogate Model

The best-performing surrogate is the one titled S4 in Table 5. From Fig. A1, we observe that all tuned NNs (S3 and S4) outperform their linear counterparts (S1 and S2) in terms of accuracy. This result is expected to a certain degree: while the linear model may not fully capture the non-linear dynamics of the high-fidelity model, we selected it to assess the extent to which a simpler model can approximate the EMS, given that many of the HF EMS model's constraints are linear. Among the linear models, model S2, which uses SVD in addition to normalization, slightly outperforms model S1, which uses only normalization. The application of SVD effectively captures key trends within the broad distribution of HPP configurations, thereby improving the accuracy of the linear model S2. However, we don't observe similar results when looking at the tuned models, S3 and S4. The NN of model S3 can make better use of all the data in the absence of SVD rather than using a reduced representation of it, as in model S4, which explains the difference between both tuned FNNs. There is a substantial difference between surrogates using only normalization (S1 and S3) and those using SVD in addition to normalization (S2 and S4). This difference is even more pronounced when comparing the tuned neural network models: model S4 (with SVD) converges in 5 hours, whereas model S3 (without SVD) takes 7 days to converge. This disparity is mainly due to the difference in training data dimensionality, as highlighted in Table 6. The use of SVD reduces the number of features, simplifying the model and accelerating the training process with very little compromise on accuracy.

Based on the results presented in both the figure and the table, we conclude that the tuned neural network using SVD—model S4—provides the best compromise between accuracy, training time, and execution time.

To gain deeper insights into the performance of model S4, we investigated its capability to approximate each normalized hourly output time series individually across all validation data. Figure 5 provides an overview of these results. From this figure, we observe that the power output of the HPP and the curtailed power are well estimated by the surrogate model. However, the battery charge and discharge profiles are more challenging to compute accurately. To further understand these discrepancies, we examine the calculated output time series for a given day from the surrogate, as well as the corresponding output time series from the high-fidelity model.

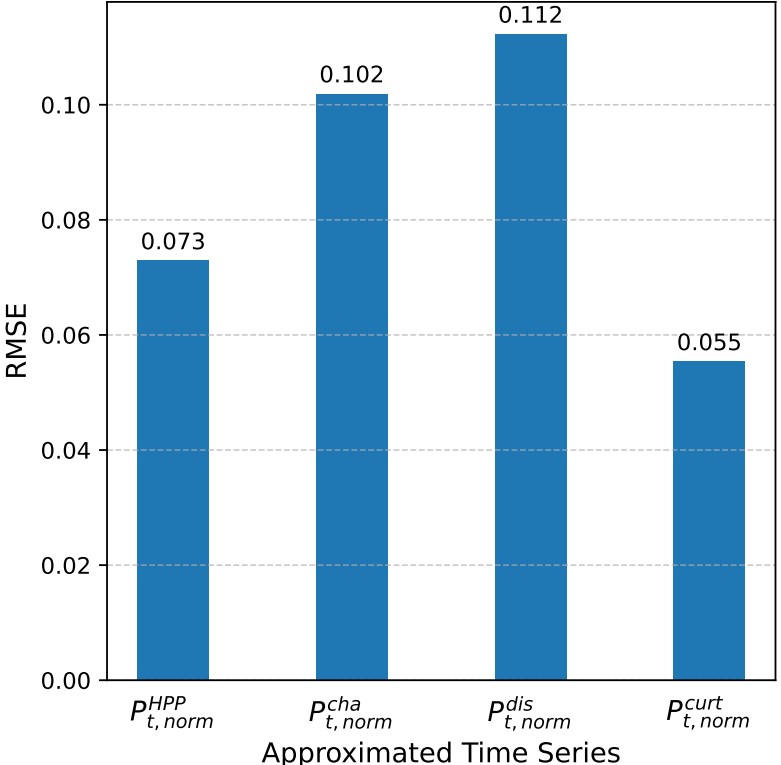

**Figure 5.** RMSE for each surrogate's output time series across all validation HPP configurations. For the definition of each variable, refer to Eq. 1 to 4

Figure 6 shows the difference between the surrogate's approximation and the HF EMS' outputs. The surrogate captures the daily trend well across all time series. While it accurately estimates the intra-day fluctuations for power bidding, it is less precise when estimating battery charge and discharge power. This is due to the abrupt power fluctuations in the high-fidelity model, as seen in Fig. 6(b) and 6(c). Additionally, as shown in Fig. 6d, the surrogate occasionally struggles to approximate consistent zero values over an entire day—a challenge characteristic of FNNs. Nonetheless, these discrepancies are minor,

with estimated curtailed power fluctuating within a ±1.5 MW range instead of the expected steady 0 MW. Such variances are

negligible relative to the HPP's export capacity, which can reach up to 700 MW.

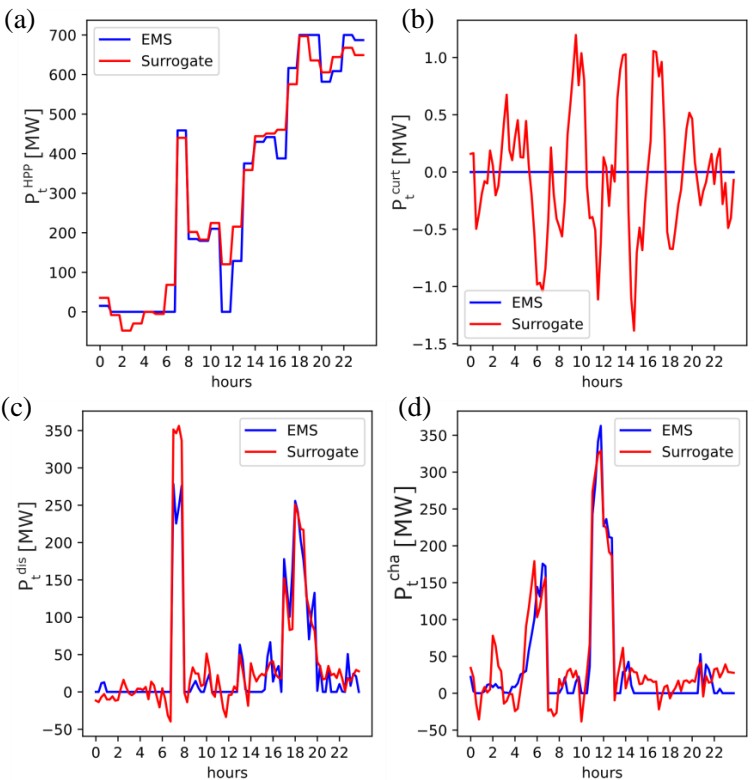

**Figure 6.** Output time series for a given day, from the high-fidelity model (blue) and the surrogate (red). All time series are in MW. (a): power output of the HPP, (b): battery charge profile, (c): battery discharge profile, (d): curtailed power

The application presented in Section 3, requires only the HPP power output out of all the calculated output time series. That is why we further examine this output time series. The left panel of Fig.7 presents a hexbin plot that compares hourly estimated, and normalized, HPP power outputs across all HPP configurations in the validation dataset ($P^{HPP}_{Surrogate}$). The hexagonal bins group nearby points (denoted as count in Fig. 7(a)) and show the density of data points within each bin. The density value

is shown on the color bar; the darker the color, the denser the hexagon. A log scale is used for clarity. A one-to-one line, representing a perfect model, is also depicted for comparison. The power bidding on the SM aligns closely with the observed values ($P^{HPP}_{EMS}$).

The PDF of errors for the same data is shown on the right-hand side of Fig. 7. The histogram (in blue) shows the frequency distribution of these errors, while the red line represents a Gaussian (normal) distribution fitted to the data. The parameters of the Gaussian fit—mean ($\mu$) and standard deviation ($\sigma$)—are shown in the legend and are 0.00 and 0.07, respectively. The

RMSE is also 0.07, indicating the typical magnitude of computation errors.

The mean ($\mu$) being equal to zero suggests that the surrogate's calculations are unbiased on average. The Gaussian fit's close alignment with the histogram suggests that the errors are distributed in a manner consistent with a normal distribution, which often implies that the surrogate model's residuals are well-behaved in a statistical sense.

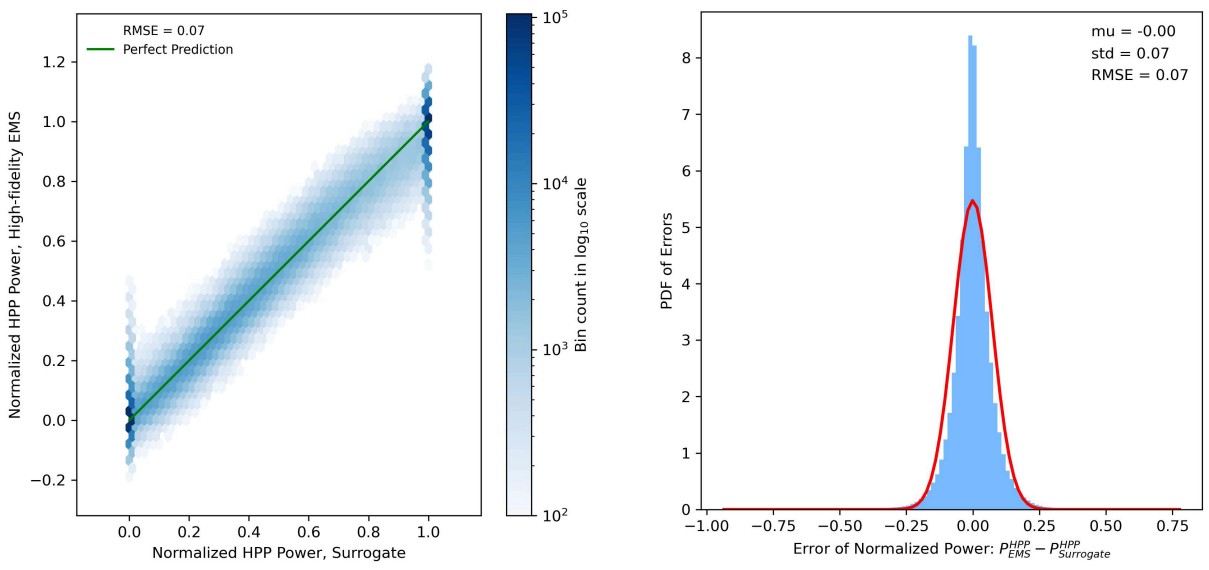

**Figure 7.** Accuracy of model S4 in approximating hourly normalized power output (see Eq. 1) for all HPP from the validation dataset. Left: the hexbin plot, group nearby points and show the density of data points within each bin, refer to the colorbar for the density of each bin. Right: PDF of error, the red line indicates a Gaussian fit, with parameters detailed in the legend.

## 5.2 Surrogate Convergence to Training Dataset

The previous study demonstrated the capacity of the NN to replicate the daily trends of the high-fidelity EMS. However, the chosen data was based on an arbitrarily large number of HPP configurations. Consequently, we sought to examine how the NN's accuracy varies with different dataset sizes. Several surrogates were trained based on model S4 with varying training dataset sizes, ranging from 4 to 200 HPP configurations. The validation dataset from the previous study is not modified to ensure a fair comparison. Results of these simulations are illustrated in Fig. 8. Interestingly, the RMSE seems to plateau when reaching a training dataset size of 110 HPP configurations. We also note that there is only a marginal gain in accuracy between 50 and 100+ HPP configurations. This is relevant because it suggests potential reductions in the data generated by the high-fidelity EMS, and, therefore, shorter training durations for the surrogate. As a reminder, each HPP configuration, which spans one year of data, requires 47 minutes to generate outputs using the high-fidelity EMS. It is also interesting to compare the Normalized Root Mean Square Error (NRMSE) of yearly revenues, computed as per Eq. 6. The model trained with 200 HPP configurations has an NRMSE of 0.81%, while the model trained with 32 HPP configurations has an NRMSE of 1.0% across

the entire validation dataset. Again, the difference between both outcomes is marginal, suggesting further reductions in training time and during the data generation process.

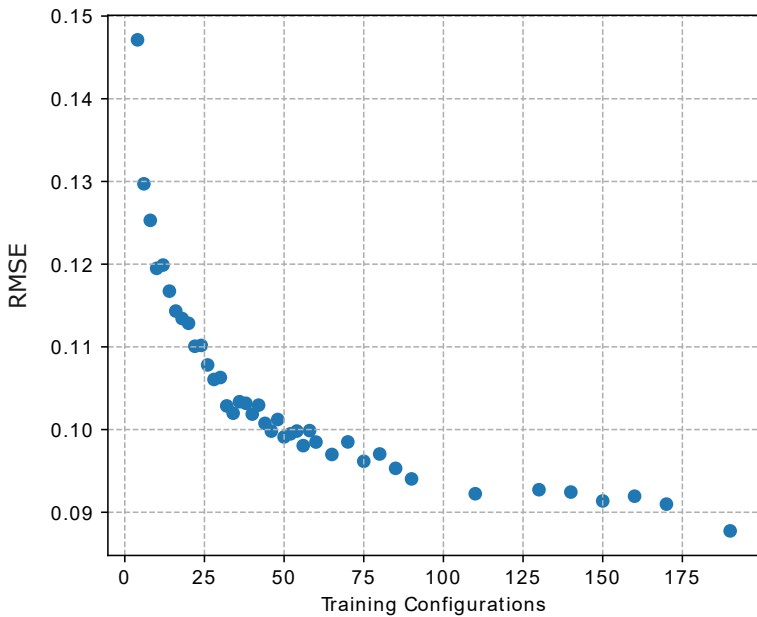

**Figure 8.** Evolution of accuracy with increasing training dataset size. A fixed validation dataset, of 50 configurations, is used across all simulations.

### 5.3 Intra-generalizability Accuracy

n this section, we evaluate the surrogate accuracy of model S4 at four different locations (A to D). Surrogate S4 is the one whose results are detailed in Section 5.1. As a reminder, this surrogate is trained using weather data from location X and with a training dataset of 200 HPP configurations. As a reminder, this surrogate is trained using weather data from location X and with a training dataset of 200 HPP configurations. We use the NRMSE of yearly revenues to measure the accuracy of the surrogate at each location. This was done on 10 randomly selected HPP configurations from the training dataset and 10 others from the

validation dataset. All accuracy results are compared to the baseline, i.e., using location X. The accuracy of the surrogate is illustrated in Fig. 9.

The surrogate model's NRMSE for computing revenue shows a marginal difference between training and validation datasets. Specifically, the NRMSE for the training dataset (location X) is 0.79%, compared to 0.81% for the validation dataset. When looking at locations A to D, the average NRMSE for the training dataset samples is 0.79% (aligning with the Train Baseline),

whereas it is 1.3% for validation dataset samples. Notably, location D shows the greatest discrepancy in NRMSE between training and validation samples. This variation may be attributed to the combination of HPP configurations and distinct weather

time series at location D, as detailed in Fig. 4. Overall, despite location X's distribution with two distinct peaks (around 0.08 and 0.1) that aren't observed in other locations, the surrogate's performance remains consistent across all locations.

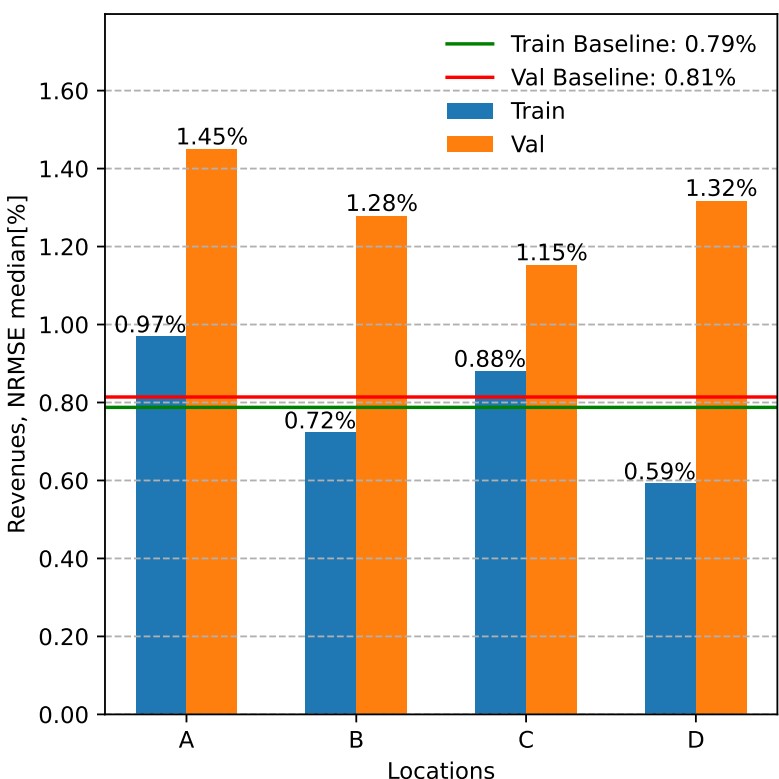

**Figure 9.** Performance of surrogate S4's generalizability for different locations and for HPP configurations from different datasets. The Baseline refers to location X, where the surrogate has been trained. Train Baseline and Val Baseline correspond to the NRMSE of Revenues from the training dataset of location X (200 HPP configurations) and from the validation dataset (50 HPP configurations). Val refers to 10 HPP configurations randomly selected from the validation dataset, while Train refers to 10 random HPP configurations from the training dataset. Both Val and Train are evaluated on locations A-D.

## 5.4    PI Evaluation Accuracy

In this section, we evaluate the PI of several HPPs using the surrogate S4 (presented in Section 5.1) and the high-fidelity EMS model. Both frameworks are described in Fig. 3. To evaluate the accuracy of the PI computed with the surrogate, we use the same 50 HPP configurations from the validation dataset for both frameworks. Figure 10 shows the PI calculated using the high-fidelity EMS on the y-axis and the PI inferred using the surrogate on the x-axis for the corresponding HPP configuration.

The RMSE of PI across the validation dataset is 0.010, indicating the average magnitude of the errors between the surrogate's estimations and the high-fidelity EMS evaluations. The scatter plot shows that most points are close to the line of perfect

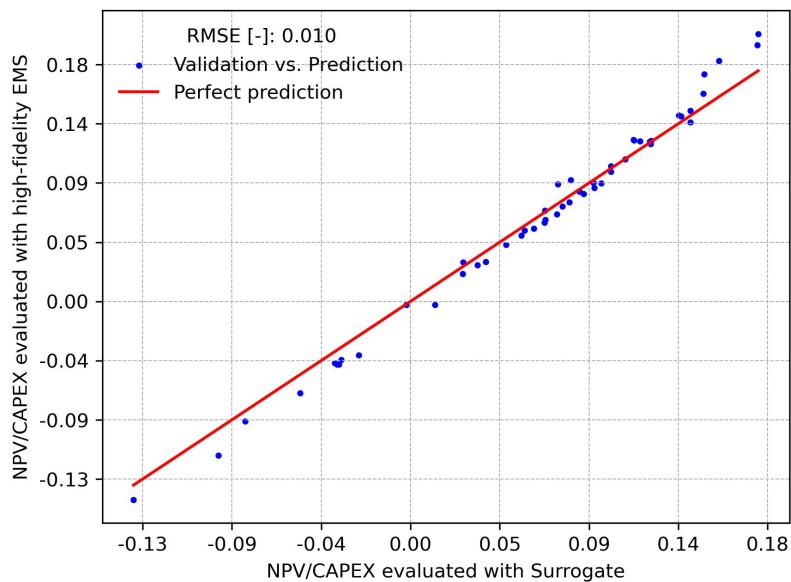

**Figure 10.** PI comparison based on surrogate inference and high-fidelity EMS evaluation for the validation dataset.

calculations (red line), with some scatter around it. Most points are below the perfect line, indicating that the surrogate slightly overestimates the profitability of the HPP. However, this tendency is reversed for higher NPV/CAPEX, where the surrogate provides a conservative estimate of the PI. Overall, the tight clustering of points around the red line suggests that the surrogate model is reliable for computing the PI when compared to the high-fidelity EMS.

## 6   Discussion

This study aims to evaluate the potential of applying surrogate modeling to emulate the behavior of a complex EMS for an HPP with bidding on the spot market. Given the increasing integration of renewable energy systems — particularly wind power — this research provides a practical approach to optimize HPP operations, enabling more efficient system integration and financial assessment.

Our investigation highlights the importance of both pre- and post-processing of data with an appropriate choice of the surrogate model. Among the options explored, the tuned FNN that utilizes SVD (surrogate S4) emerged as the optimal balance. Indeed, Fig. A1 shows that the tuned NN using only normalization (surrogate S3) is the most accurate, while adding SVD (surrogate S4) results in similar performance. Yet, when we look at the computational time, as shown in Table A1, using SVD (in surrogates S2 and S4) significantly reduces the training duration. This difference is even more pronounced when the tuning time is considered: it requires five days to tune the NN using normalization (surrogate S3), whereas it takes 4.3 hours when SVD (surrogate S4) is used. This discrepancy is attributed to the inherent capability of SVD to extract a reduced order of data that contains meaningful coefficients and daily temporal trends.

However, challenges persist, particularly in estimating battery charge and discharge profiles. As depicted in Fig. 5 and 6, this difficulty arises from the high-fidelity model's abrupt power output fluctuations and the intrinsic non-linearity of these variables. In particular, Fig. 6(c) and (d) highlight the surrogate's limitations in accurately and consistently representing the battery's hourly operation. In several time steps, the model shows the battery charging and discharging simultaneously, and at times, the charge and discharge power take on negative values, which is physically unrealistic. These issues stem from the nature of the regressor; the FNN cannot inherently capture the physical constraints that would typically be enforced by multiple equations. Specifically, the FNN lacks explicit equations to govern its outputs. However, for the purposes of this study, focusing primarily on the power output of the HPP is sufficient, as this is the only variable required in revenue calculation and subsequent PI evaluation. Additionally, the power output of the HPP is well estimated, as shown in the PDF plot of Fig. 7: there is no bias, and the standard deviation is very small. While this result only holds for a surrogate trained with 200 HPP configurations, it is still reasonable to assume similar behavior for a surrogate trained with fewer data points. Indeed, Fig. 8 demonstrates a marginal difference in accuracy between a surrogate trained with 200 HPP configurations and one trained with 50 HPP configurations. Ultimately, it is a trade-off between training time and accuracy. In terms of intra-regional generalizability, the accuracy across different locations is more consistent than between dataset types. This uniformity in accuracy within each dataset type can be partly attributed to the region's relatively homogeneous wind profiles, facilitated by its largely flat terrain. A loss in accuracy is observed when unseen HPP configurations are used (e.g., validation dataset). Nonetheless, these results demonstrate the surrogate's ability to capture essential data trends (Fig. 9). However, it is important to note that this study's scope was confined to the DK1 market region, characterized by uniform wind profiles due to its flat terrain (Fig. 4). The fast and accurate surrogate allows us to evaluate an HPP's profitability throughout its lifetime with little computational burden. Indeed, the surrogate model is capable of evaluating the NPV/CAPEX for all 50 HPP configurations in Fig. 10 in 25 seconds. In contrast, computing the same evaluations using the high-fidelity model for each HPP configuration, with inputs spanning over a year, would take approximately 39 hours. However, it is important to understand the impact of the surrogate's accuracy on the PI. Figure 10 shows that the surrogate can be reliably used if slight deviations of around 0.010 in the PI are acceptable for the intended business evaluation. In other words, the error on the computed NPV is around 1% of the CAPEX. It is also relevant to note that not all HPP configurations are profitable, resulting in negative PI, further supporting the use of NPV/CAPEX as an evaluation metric. Hence, the importance of optimization in the context of HPP sizing, which is enabled with the developed framework. However, a detailed exploration of the sizing optimizer is beyond the scope of this manuscript and will be the subject of future investigations. It is important to emphasize that these findings are site-specific and heavily dependent on the cost model employed; hence, they should not be generalized across different HPP sites.

There are certain limitations and future works worth acknowledging. For one, the full capabilities of the high-fidelity model have not been leveraged. While the EMS can consider a realization of the forecast error in both weather and market data, our initial approach prioritized a methodology using perfect forecast data. Nonetheless, this is a natural next step, where a sizing framework can be developed based on a surrogate that can handle the inherent uncertainties in weather and market forecast errors. While our research mainly focused on the spot market, currently the major source of revenues for power plants, market dynamics might shift. As the share of intermittent power plants increases in the grid system, which is becoming more

decentralized, the balancing market is forecasted to become a considerable revenue stream. Thus, there is a pressing need for a more comprehensive surrogate considering operational strategies in both spot and balancing markets. Moreover, an FNN has its own limitations when it comes to time series representation, as seen in the battery charge and discharge profiles in Fig. 6. This highlights the importance of further exploring the machine learning field. A promising avenue would be models that blend physical constraints, such as physics-informed neural networks. Additionally, to develop a more robust sizing methodology, it is necessary to account for various forecast scenarios for wind generation and market prices within the EMS. This can be achieved through methods from the field of surrogate-based optimization under uncertainty. These considerations will be addressed in future work.

## 7 Conclusion

In this paper, we have introduced a new methodology to accurately and efficiently approximate a state-of-the-art EMS for HPPs involved in spot market power bidding. This model leverages singular value decomposition to extract temporal trends in the input and utilizes an FNN to represent the non-linear dynamics of the EMS. This method has demonstrated over twice the accuracy of traditional multivariate linear regression models. A key innovation of our study is the combined use of SVD and FNN, which represents a novel approach in this field. This approach successfully replicates the annual revenues of an HPP with an NRMSE of 0.81% for the best model. To fully demonstrate the capabilities of our surrogate model, we have integrated it into a sizing evaluation framework designed to calculate the Profitability Index ($NPV/CAPEX$) based on the technology mix rating within the HPP. This framework not only enabled substantial computational savings—reducing processing time from 39 hours to 25 seconds compared to a high-fidelity model—but also achieved an RMSE of 0.010. Although our methodology is straightforward, it is nonetheless powerful and opens up new possibilities for optimizing HPP sizing in the context of renewable energy integration. This study emphasizes the relevance of surrogate modeling to the wind energy field, where efficient and accurate tools are essential for navigating the increasing complexity of renewable energy markets and supporting the transition toward sustainable energy systems.

*Data availability.* The weather and spot market price time series data are available on request from the corresponding author.

## Appendix A: Surrogate Models Comparison

The accuracy of the four models, presented in Table 5, can be found in Fig. A1. The RMSE of all normalized hourly output time series is used to compare the accuracy of the models. Moreover, the training and inference times are reported in Table A1.

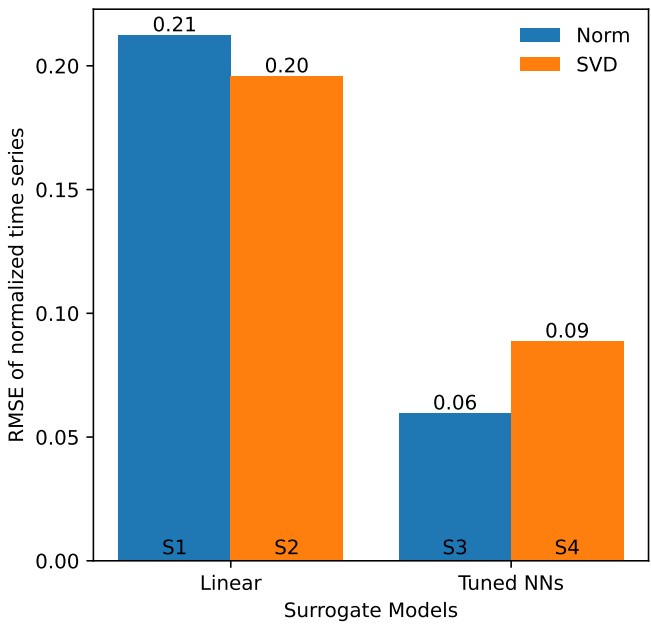

**Figure A1.** Validation RMSE by Data Surrogate model: "Norm" for models using a normalization only, and "SVD" for models using a combined normalization and SVD.

Table A1 compares the time needed to execute the methodology for each surrogate model. The pre-processing (Pre-proc.) time considers both training and validation datasets. However, the training time (Train Time) accounts only for the training dataset, while the inference time (Inf. Time) reflects the inference on a single HPP configuration spanning one year of data.

**Table A1.** Time Metrics of Surrogate Models

| Model Name | Pre-proc. | Train Time | Inf. Time |
|---|---|---|---|
| S1 | 1.1m | 14.4h | 0.64s |
| S2 | 7m | 14m | 0.02s |
| S3 | 1.1m | 7d | 1.02s |
| S4 | 7m | 5h | 0.04s |

## Appendix B: FNN architecture

Tables with tuned FNN architecture.

**Table B1.** Architecture of model S4

| Layers | Neurons |
|---|---|
| Input Layer | 17 |
| Hidden Layer 1 | 80 |
| Hidden Layer 2 | 60 |
| Hidden Layer 3 | 80 |
| Hidden Layer 4 | 80 |
| Hidden Layer 5 | 80 |
| Hidden Layer 6 | 80 |
| Hidden Layer 7 | 60 |
| Hidden Layer 8 | 80 |
| Hidden Layer 9 | 80 |
| Output Layer | 125 |

**Table B2.** Architecture of model S3

| Layers | Neurons |
|---|---|
| Input Layer | 5 |
| Hidden Layer 1 | 80 |
| Hidden Layer 2 | 80 |
| Hidden Layer 3 | 80 |
| Hidden Layer 4 | 80 |
| Output Layer | 4 |

## Appendix C: Cost Model

The cost model is described in this section. The CAPEX depends on $C_w$, $C_b$, and $C_{el}$, which are the CAPEX of the wind power plant, batteries, and the Balance of System (BOS). Similarly, the OPEX is the sum of $O_{w,y}$, $O_{b,y}$, and $O_{el,y}$, which are the yearly OPEX from the wind power plant, batteries, and BOS. $C_w$ is proportional to the wind turbine's cost ($WT_{cost}$) and the cost of civil works ($WT_{civil}$) in $EUro/MW$. Meanwhile, $C_{el}$ is proportional to a combination of the number of

battery equivalents in today's value ($Nb_{eq}$), the battery energy cost per MWh ($B^E_{cost}$), the battery power cost ($B^P_{cost}$), civil costs ($B^P_{civil}$), and control system costs ($B^P_{control}$) per MW. $C_{el}$ depends on the shared BOS cost ($C_{BOS}$), the grid connection cost of the HPP ($P^G_{cost}$), and the cost of land in $Euro/km^2$ ($L_{rent}$). The OPEX of the wind power plant is calculated based on the fixed and variable Operation and Maintenance (O&M) costs of the wind turbine per year and per MW ($WT^{OM}_{fixed,y}$ and $WT^{OM}_{variable,y}$), as well as the mean Annual Energy Production (AEP) of the wind power plant ($mean(AEP)$). Meanwhile, the battery's yearly OPEX is proportional to the yearly O&M cost of the battery per MWh ($B^{E,OM}_y$). The equations are as follows:

$$CAPEX = C_w + C_b + C_{el}$$

$$OPEX_y = O_{w,y} + O_{b,y} + O_{el,y}$$

$$C_w = (WT_{cost} + WT_{civil}) \cdot P^W$$

$$C_b = Nb_{eq} \cdot B^E_{cost} \cdot B^E + (B^P_{cost} + B^P_{civil} + B^P_{control}) \cdot B^P$$

$$C_{el} = (C_{BOS} + P^G_{cost}) \cdot P^G + L_{rent}$$

$$O_{w,y} = WT^{OM}_{fixed,y} \cdot P^W + mean(AEP) \cdot WT^{OM}_{variable,y}$$

$$O_{b,y} = B^{E,OM}_y \cdot B^E$$

$$O_{el,y} = 0$$

In this study, we set a fixed lifetime for the battery ($i_b$) as battery degradation is not considered. Additionally, to address the decreasing costs of batteries over time, we employ the concept of the equivalent number of present batteries ($Nb_{eq}$). This method incorporates the annual battery price reduction rate ($f_b$) and the designated replacement year for each battery ($y_b(i_b)$).

$$Nb_{eq} = \sum_{i_b=0}^{N_b-1} (1 - f_b)^{y_b(i_b)}$$

**Appendix D: Data supplement**

Table with the coordinates of each location used in the intra-generalizability study.

**Table D1.** Location Coordinates for the Generalizability study. Coordinates are shown in decimal degrees.

| Location | Latitude | Longitude |
|----------|----------|-----------|
| X | 57.0482 | 8.8876 |
| A | 56.383 | 8.6705 |
| B | 55.2908 | 8.6551 |
| C | 57.1852 | 9.9527 |
| D | 55.3088 | 10.4398 |

Table D2 presents a summary of the cost assumptions used in this article. As the battery's lifetime is seven years, each HPP
will require three batteries during its lifetime. Given a battery price reduction of 10% per year, we obtain an equivalent number
of batteries ($Nb_{eq}$) of 1.84.

**Table D2.** Cost assumptions

| Variable | Value |
|---|---|
| $r_{AT}$ | 6% |
| $\tau_{tax}$ | 22% |
| $WT_{cost}$ [MEUR/MW] | 0.46 |
| $WT_{civil}$ [MEUR/MW] | 0.19 |
| $WT^{OM}_{fixed,y}$ [MEUR/MW/year] | 9,000 |
| $WT^{OM}_{variable,y}$ [EUR/MWh/year] | 0.97 |
| $B^{E}_{cost}$ [EUR/MWh] | 90,000 |
| $B^{P}_{cost}$ [EUR/MW] | 32,000 |
| $B^{P}_{civil}$ [EUR/MW] | 36,000 |
| $B^{P}_{control}$ [EUR/MW] | 9,000 |
| $B^{E,OM}_{y}$ [EUR/MWh/year] | 0 |
| $f\_b$ | 10% |
| $i\_b$, lifetime of battery [years] | 7 |
| $Y$, lifetime of HPP | 25 |
| $C_{BOS}$ [MEUR/MW] | 0.12 |
| $P^{G}_{cost}$ [MEUR/MW] | 0.05 |

*Author contributions.* C.A., J.P.M.L. and K.D., conceived the research, C.A., J.P.M.L., and J.Q., developed the methodology C.A. and J.P.M.L. developed the code for the surrogate and gathered the required data, C.A., J.P.M.L., J.Q., and T.G., performed the formal analysis, J.P.M.L., K.D., S.G., and T.G. supervised the work, C.A., J.P.M.L., and J.Q., wrote the original draft and J.P.M.L., K.D., S.G., and T.G
reviewed and suggested modifications for the draft.

*Competing interests.* The authors declare that they have no conflict of interest.

*Disclaimer.* This work is funded by TotalEnergies under the project "Hybrid Power Plant Life-cycle Optimization".

*Acknowledgements.* We would like to thank Juan-Andrés Pérez-Rúa and Poul Ejnar Sørensen for their critical feedback on the methodology and the figures as well Jenna Iori for her feedback after the first draft was written, and finally Rujie Zhu for sharing the high-fidelity EMS script.

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
