# Peer review of "Enabling Efficient Sizing of Hybrid Power Plants: A Surrogate-Based Approach to Energy Management System Modeling"

_Wind Energy Science, 2024_

## Author Comment (AC1)

**Reply on RC1 – PDF Format**

- **Detailed Comment 1:**
Figure 8 reports the output time series for the high-fidelity model and for the surrogate model. It can be seen on the figure that at t=0 h the charge power is positive and the discharge power is negative. One would expect the discharge power to be zero. Furthermore, the power output of the HPP is negative between t=2 and 4 h, assuming that the HPP is able to charge from the grid. These two points raise questions on the ability of the surrogate model to represent a valid storage model. Please expand the discussion of Figure 8 to include these considerations.

Answer 1:
The purpose of showing the time series in Figure 8 (now Figure 6) is to demonstrate the hourly performance of the surrogate. While the trends and fluctuations within the HPP output are well captured, the same cannot be said for the other time series. This highlights the ability of the surrogate to represent the HPP operation as a whole, but its inability to describe the hourly operation of each technology individually with high accuracy. These issues arise from the nature of the regressor; the FNN cannot inherently capture the physical constraints that multiple equations would typically enforce. Specifically, the FNN lacks explicit equations to govern its outputs. However, for this study, focusing primarily on the power output of the HPP is sufficient, as this is the only variable required in revenue calculation and subsequent profitability index evaluation.

Changes in paper:
Additional discussion is added in Section 6 to elaborate on that. l. 529-537

- **Detailed Comment 2:**
The story line of the paper focuses on the field of system integration of renewable energy and hybrid power plants. In order to fit better in the scope of Wind Energy Science, consider highlighting the relevance of this study for the field of wind energy in the introduction, discussion and conclusion of the manuscript.

Answer:
Additional text was added at the start of the introduction and discussion and at the end of the conclusion to highlight the study's relevance.

- **Detailed Comment 3: Motivation and research question:**

3.1:
l. 37: " The importance of a realistic EMS …" : it is unclear from the literature review what is the difference between a high- and low-fidelity model for the EMS. The manuscript provides an overview of different high-fidelity models used in the field but does not describe the low-fidelity ones. The following reference may be of interest in this context:
Stanley, A. P. J., & King, J. (2022). Optimizing the physical design and layout of a resilient wind, solar, and storage hybrid power plant. Applied Energy, 317, 119139.
https://doi.org/10.1016/j.apenergy.2022.119139

Answer:
To explain the differences between the High-Fidelity (HF) and LF (EMS), additional literature was added within the introduction (l. 59-76), and a new table was added: Table 1. Comparison of HF and LF EMS Models.

3.2:
Table 1 reports that 1000 "iterations [are] required to find [a] refined solution". Please put this number in context with the literature. Is it characteristic of HPP sizing problems to require a large number of iterations to converge?

Answer:
The short answer is that from previous work that authors are familiar with (Leon et al., 2024), we need to evaluate hundreds of different sizes to reach an optimal sizing.

Changes in Paper:
The previous Table 1 is now removed and replaced with another table, Table 1. Comparison of HF and LF EMS Models. Instead, we provide more details on the sizing problem in the introduction l. 76-102.

3.3:
l. 59: "HPP sizing optimization often relies on a simplified EMS representation": the introduction of the paper does not describe the problems associated with using a simplified or low-fidelity EMS model. As such, it is unclear why one would prefer a high-fidelity model in a context where computational cost is a problem. Please describe explicitly and quantitatively why a high-fidelity model is superior to a low-fidelity one.

Answer:
Although no studies directly compare HF and LF EMS models, it is evident that LF EMS models sacrifice accuracy due to simplifications in areas such as component modeling, market bidding strategies, and operational constraints. These simplifications can result in inaccurate power schedules, leading to revenue projections that may misrepresent the business case for a given HPP sizing. Quantifying the extent of deviation between low- and high-fidelity EMS models is beyond the scope of this paper; however, we are currently researching this topic.

Nonetheless, Zhu et al. 2024 examined the accuracy of total profits across three HF EMS models, demonstrating that even among HF models, certain simplifications commonly found in LF models—such as relying on deterministic forecasts—can lead to revenue discrepancies of up to 7.6% when compared to the best-performing model (refer to Table 3 of the paper).

Although this provides some insight into the potential impact of LF EMS simplifications, a comprehensive study comparing HF and LF EMS models for HPPs is currently being conducted by colleagues in our research group.

Changes in paper:
A paragraph (l. 103-109) was added to explain the current state of the research on comparing HF and LF models:

3.4:
The list of major contributions is a good addition to the introduction. Consider writing explicitly the research question for the study.

Answer:
The research question was added before the list of contributions. l.166-167

3.5:

l. 100-101: "Integration of the developed surrogate within a framework to evaluate the profitability of an HPP sizing with high accuracy.": this statement is convoluted. Consider combining it with the previous one, e.g.: "Assessment of the surrogate model's ability to calculate … time series and profitability of the HPP …"

Answer:
As the three points are distinct, we would like to keep them separate. The wording was modified to make it more straightforward.

Changes in Papers:
The second and third points were reformulated (l.171-174).

- Detailed Comment 4: Methodology
4.1:
Two of which use a multivariate linear regression to establish a baseline and two others are based on Neural networks (NNs)": please put in context the choice of surrogate models in the introduction. What are the strengths and weaknesses of these models? Have they been applied to models similar to HPP dispatch strategies? Can one expect them to perform well for this type of problem? Are there other surrogate models one could consider for HPP dispatch strategies?

Answer:
In the revised introduction, we have discussed data-driven surrogate models, showcasing successful applications of both regression models—ranging from linear to more complex Neural Networks (NNs). These models are often applied to solve problems involving partial differential equations where a large number of parameterized instances must be evaluated. In such cases, thousands of degrees of freedom are typically required to achieve accurate solutions, leading to significant computational demands, especially in scenarios requiring real-time simulations.
This challenge is similar to our current problem, where the high-fidelity EMS model must be solved across hundreds of HPP sizing configurations. For each sizing, the EMS model operates at an order of magnitude involving hundreds of thousands of degrees of freedom, creating a substantial computational burden. Given that the use of surrogate models for EMS in grid-connected HPPs is largely unexplored in the literature, we examined a range of potential surrogate models. Our exploration includes simple approaches, such as linear regression, and more sophisticated models, like feedforward neural networks (FNNs), initially chosen based on their adaptability to our specific problem.
The linear regression model serves a dual purpose: firstly, to assess if a linear approximation can capture the essential dynamics of our problem, and secondly, to provide a baseline against which we can measure the improvement in accuracy and computational cost when using more complex surrogate models.

Changes in paper:
Additional text was added: l.120-149

4.2:
In the first part of section 5, the four surrogate models are compared. However, only one model is used for the rest of the results section. This leads to a lengthy result section, where the high impact results are more difficult to identify. Consider either (i) focusing on the best surrogate model in the entirety of section 5 and moving the comparison between linear and NN models in an appendix; or (ii) compare all four models for all relevant metrics. This second approach would help the reader have insight on the trade-off between accuracy and computational time.

Answer:
Thank you for the suggestion. We decided to go with the first option, and changes were made in Section 5 accordingly.

Changes in paper:
Figure 6 is now Figure A1, and Table 10 is now Table A1.
Moved previously numbered Figure 6 (now Figure A1) and Table 10 (now Table A1) to Appendix A: Surrogate Models Comparison and the corresponding text.
Only figures related to the best-performing surrogate models (S4) were kept in Section 5.

4.3:
The surrogate models are compared to each other, with the high-fidelity EMS as a reference. However, the results would be significantly strengthened if the comparison were to include a low-fidelity model. For example, if the RMSE for the low-fidelity EMS was 10 times higher than the linear and NN models (Figure 6), this would provide an excellent motivation for the use of a surrogate.

Answer:
Thank you for the suggestion. While I agree that including a comparison with a low-fidelity EMS would provide additional context and further strengthen the results, the focus of this study is specifically on the use of surrogate models as a computationally efficient alternative to high-fidelity EMS. As Zhu et al. (2024) demonstrated, even minor simplifications typical of low-fidelity EMS can lead to significant revenue discrepancies, with observed differences reaching up to 7.6% compared to the most accurate high-fidelity model. This suggests that the discrepancy would be even greater with a low-fidelity EMS.

Additionally, our colleagues' ongoing research explicitly addresses the comparison between high- and low-fidelity EMS models. While the insights from that work would be valuable here, incorporating it is beyond the current scope of this paper. Instead, this study focuses on evaluating the accuracy and computational benefits of surrogate models relative to high-fidelity EMS, as this comparison more directly aligns with the primary goals of the research.

4.4:
l. 208: "applied for the 2nd and 4th surrogate models": consider introducing a name for each surrogate, instead of referring to their number in Table 5. The names should match the labels of the figures in the results section.

Answer:
The names of the surrogates have been changed to S1-S4, please refer to Table 5 for the definition of each model. Other changes in the text and figures have been applied.

4.5:
l. 407: "as the linear model cannot capture the inherent non-linearities of the high-fidelity model.": why has a linear model been chosen? This statement suggests that the choice of methodology is not appropriate for the study.

Answer:
Thank you for pointing that out. I will revise the statement to clarify this aspect.

In addition to the reasons mentioned earlier for selecting the linear regression model, we recognize that the EMS exhibits some non-linear behaviors. However, we also suspect these non-linearities are relatively mild, as most of the system's constraints are linear. Therefore, we included the linear regression model to evaluate its effectiveness in approximating the high-fidelity model. This approach allows us to establish a baseline and assess the extent to which a simple model can capture the EMS's behavior before moving on to more complex surrogate models.

Changes in paper:
Sentence added in l.437-439

- Detailed Comment 5: Literature review
5.1:
On the topic of energy markets and subsidies for renewable energy, consider referring to the following report: European University Institute: Robert Schuman Centre for Advanced Studies, Kitzing, L., Held, A., Gephart, M., Wagner, F., Anatolitis, V., & Klessmann, C. (2024). Contracts-for-difference to support renewable energy technologies : considerations for design and implementation, European University Institute. https://data.europa.eu/doi/10.2870/379508

Answer:
Thank you for the recommendation. The paper is now cited in the introduction, and additional context was added since the writing of this report, e.g., the Agreement of May 2024. l.20-24

5.2:
The literature review would be strengthened by citing literature related to machine learning and data-driven methods. An overview of methods for modelling time-series would be a good addition to the paper.

Answer:
Additional literature on these topics has been added. l.120-149

5.3:
Consider including a short description of the advances done on hybrid power systems (e.g. for micro-grids). This would help contextualize better the paper since the problem of storage sizing and dispatch strategy has been addressed in this field before, even though not in relation to electricity markets.

Answer:
Additional context was added in l.53-58.

5.4:
The literature review could be complemented by citing articles related to the field of bidding strategies. For example, the works by Pierre Pinson or Kenneth Bruninx may be of interest: Ding, H., Pinson, P., Hu, Z., & Song, Y. (2016). Integrated Bidding and Operating Strategies for Wind-Storage Systems. IEEE Transactions on Sustainable Energy, 7(1), 163–172. https://doi.org/10.1109/TSTE.2015.2472576
Toubeau, J.-F., Bottieau, J., De Greeve, Z., Vallee, F., & Bruninx, K. (2021). Data-Driven Scheduling of Energy Storage in Day-Ahead Energy and Reserve Markets With Probabilistic Guarantees on Real-Time Delivery. IEEE Transactions on Power Systems, 36(4), 2815–2828. https://doi.org/10.1109/TPWRS.2020.3046710

Answer:

Thank you for the suggestion. These papers have been cited.

- Detailed Comment 6: Structure of the paper

6.1:
The last sentence of the first paragraph states that BESS are valuable to establish robust business cases. Then, the second paragraph discuss the definition of HPPs. In this case, the link between the two paragraph is not clear. Instead, consider introducing HPP in the first paragraph, and narrowing the focus of the study on HPP with storage systems.

Answer:
Both paragraphs were modified according to the comment: l. 27-36

6.2:
l. 68- 71: "To evaluate the value of HPP, …": this paragraph discussing performance metrics for HPPs does not seem relevant in the introduction. Consider moving it in a later section describing the profitability index.

Answer:
Agreed. This sentence was integrated in Section 3 (l. 349-353), where financial metrics are discussed.

6.3
l. 118-125: "In electricity trading …": the description of the spot and balancing market does not help describing the methodology of the study. Consider moving this paragraph to the introduction.

Answer:
The paragraph was shortened and integrated in the introduction: l. 41-52.

6.4
Consider restructuring section 2 and 3 into one section describing the metrics relevant to HPPs (including the description of the relevant time series, the EMS and the profitability index) and one section describing the surrogate models.

Answer:
After discussing this with most co-authors, the majority wished to keep the current structure.

6.5
l. 185-186 "Details on the normalization process appear later on" and l. 189 "The specific use of this method is detailed in this section": instead of referring the reader to a later part of the paper, consider restructuring the subsection.
Consider shortening the description of the normalization steps (l.194-205) and instead state that scaling is used for all time series.

Answer:
The subsection was slightly restructured, and the description was shortened as suggested.

6.6
l. 213-218: the description of the shapes of the matrices does not seem relevant for the study. Consider removing the associated sentences or moving them to an appendix.

Answer
These descriptions were removed.

6.7
Please restructure and shorten section 2.2.3. The text mixes general statements about training neural networks, mentions of the steps of the methodology (l. 235 and l. 255 "the best-performing model … is selected") and descriptions of the training methodology. This makes the subsection difficult to follow.

Answer:
It seems there is some confusion. The selection of the best-performing model is part of the training process. This does not refer to selecting the best-performing model among the four types of surrogates, S1-S4; instead, this refers to one surrogate type, e.g., S3 or S4. In the tunning process, several hundred models are evaluated for a given type of surrogate (S3 or S4). These hundreds of models differ in the choice of hyperparameters. Among these, the best performing model is selected and further trained "till convergence." The training method described needs to be applied individually for surrogate types S3 and S4.
A workflow example for model S3 would be as follows:
Apply normalization to inputs and output --> define FNN architecture and hyperparameter search space --> Proceed with tuning --> Result: hundreds of FNN trained --> Get the most accurate FNN based on the defined loss function (MSE) --> Further train that model till convergence --> Result: model S3.

The section was modified and shortened to make it more straightforward.

Please let us know if we have misunderstood your comment.

6.8
Consider moving the description of the cost model to an appendix (l. 322-344)."

Answer:
The description of the cost model has now been moved to Appendix C: Cost Model. Moreover, the data related to the cost model has been moved; previously, Table 9, now Table D2, has been moved to Appendix D: Data Supplement.

- Detailed Comment 7: Clarity and conciseness

7.1
l. 25 "power plants that combine several technologies": please precise the type of technology. Consider using the terms "electricity generation and storage technologies".

Answer:
The wording was changed to "combine several generation technologies, including wind turbines, and possibly energy storage": l.30-31

7.2
l. 32-35: "As HPPs transition to market-driven revenue models… throughout the power plant's lifetime": the start of this paragraph is vague. What are the "new possibilities and challenges" mentioned? What does the expression "navigate energy markets" mean in the context of the study? What are the characteristics of a "detailed operational strategies"?

Answer:
A paragraph was added before explaining the meaning of market-driven models, i.e., CFD. The text was further modified to explain the opportunities, challenges, and detailed operational strategies. l. 37-52

7.3
l. 35 "Energy Management System": please define the term, and highlight the difference between other types of "control" in the context of HPP. Consider stating the difference between EMS and adjacent terms such as bidding or dispatch strategies.

Answer:
A first definition of the EMS is given in the introduction l. 38-41. A more detailed definition is now given in Section 2.1 to clarify which EMS is used in this article. l. 191-200.

7.4
l. 95: "Development of a fast and precise surrogate": the term "accurate" seems more pertinent in this context.

Answer:
The term was modified.

7.5
l. 98 "Assessment of the surrogate's ability to predict hourly operational time series": consider using the verb "compute", "calculate", "model" or "estimate" instead of "predict", since the latter implies a focus on future (and unknown) data.

Answer:
Thank you for the suggestion. Similar changes were applied to the paper.

7.6
Please precise what the term "surrogate model" means in the context of the study. By itself, "surrogate" implies a simplified or approximation model, and does not refer to data-driven or machine learning methods specifically.

Answer:
Additional literature on data-driven surrogate models was added to the introduction to give context to the developed surrogate models. Additionally, the paragraph of section 2.2 gives a detailed definition of the term in this study.

7.7
l. 404: "This RMSE provides a holistic measure of the model's accuracy": why is the term "holistic" used here? Consider rephrasing.

Answer:
The sentence was changed to: "Since this RMSE is calculated across all output time series, it provides a broad assessment of the model's accuracy, without specific insights into each individual series." l.322-323.

7.8

l. 413: "Table 10 contrasts the time required to execute the workflow for each surrogate model" Consider rephrasing this sentence.

Answer:
The sentence was changed to: "Table A1 compares the time needed to execute the methodology for each surrogate model." l.592.

7.9
l. 426: "Figure 8 shows the difference between the surrogate's prediction and the ideal behavior": what does "ideal behavior" mean here? Consider rephrasing.

Answer:
The sentence was changed to: "Figure 6 shows the difference between the surrogate's approximation and the HF EMS' outputs" l. 458.

7.10
l. 537 "the synergistic use of SVD and FFN": what does "synergistic" mean in the context of the study? Consider rephrasing.

Answer:
Changed to: "A key innovation of our study is the combined use of SVD and FNN, which represent a novel approach in this field." l. 576-577.

7.11
l. 542: "a mere 25 seconds" and "remarkable accuracy": Please avoid subjective terminology and use neutral language instead."

Answer:
Removed these terminologies

- Detailed Comment 8: Figures

    8.1:
    What is the information conveyed by Figure 4? Consider removing it.

    Answer:
    The Figure is removed.

    8.2:
    Figure 5.b. : this representation of the wind distribution is unusual. A more standard representation as the probability distribution function would be more meaningful for the reader.

    Answer:
    The previously numbered Figure 5 is now Figure 4.
    Thank you for the feedback. Although the violin plot representation may be less conventional, it was chosen for its ability to convey detailed insights into the distribution of wind power across multiple locations within a single, compact visualization. Overlaid PDF plots were considered; however, they tended to appear cluttered, making it difficult for readers to extract meaningful information. Separate PDF plots for each location were also examined, but they would have required significantly more space. While a CDF plot was another option, its interpretation is less intuitive than the violin plot, especially for readers less familiar with cumulative distributions. To

enhance clarity, we have included additional text ( l. 413-426) explaining how to interpret the violin plot, which should aid in understanding its unique presentation.

8.3:
"Please follow the journal guidelines for the captions: https://www.wind-energy-science.net/submission.html#figurestables "

Answer:
Figures were changed to comply with the guidelines.

8.4:
Consider using intelligible notation in the legend and labels when possible, instead of introducing the notation in the caption. For example, the labels of Figure 7 do not correspond to previously introduced notation.

Answer:
Note that Figure 7 is now Figure 5.
The notations are now introduced in the texts and then used in the figures. The notations are now consistent among all figures, tables, and text.

8.5:
Figure 6,7 and 10: including the equation for the RMSE in the label seems unnecessary since the notation and equations is introduced in the main text.

Answer:
Figures were modified.
Note that Figures 6, 7, and 10 are now A1, 5, and 8.

8.6:
Figure 8: Please indicate the unit on the figure labels.

Answer:
The units are now included.

8.7:
Figure 9: it is unclear why two figures are relevant here. Consider removing Figure 9 (a).

Answer:
Note that Figure 9 is now Figure 7.
Figure 7(a) is shown because it is hard to visualize the extent of the scatter from the PDF plot shown in Figure 7(b). This scatter helps to understand the deviations from the HF EMS, as shown in Figure 6 (previously Figure 8).

8.8:
l. 443: "The mean (μ) being close to zero suggests…": Note that Figure 9(b) indicates that the mean is equal to zero.

Answer:
Text changed to avoid confusion: l. 475.

8.7:
Section 2.1.: a figure illustrating the EMS would be relevant to support its description in the text. For example, Figure 1 could be extended to describe the time schedule for bidding and dispatch decisions.

Answer:
Figure 1 was modified. Additional information was added to the bidding process. Additional text was added to explain further the bidding process of the EMS (l. 207-217). For clarity, throughout the article, the use of the acronym "EMS" was slightly changed: when applicable, "EMS" was modified to "SM optimization", referring to the bidding process happening at the day-ahead stage. The acronym "PMS" was removed entirely and replaced by Real-Time (RT) dispatch. The HF EMS combines both SM optimization and RT dispatch.

Detailed Comment 9: Equations

9.1:
For the presentation of the equations in the manuscript, consider introducing the relevant metrics and their notations before the equation.
Consider adding a paragraph or a subsection to introduce the notation used in the paper, since there is a wide variety of symbols, subscripts and superscripts in the manuscript.
l. 268-271: please introduce notation in a paragraph and not as a list. This comment applies to subsequent equations as well.

Answer:
Thank you for the feedback. The notations are now introduced in a paragraph before the equations.

9.2:
Equations 1 to 3 are not equations since they don't include an equal sign. Consider giving each scalar parameters a name, a symbol and describe their meaning.

Answer:
Indeed, apologies for the oversight. They are now included in the paragraph instead.

9.3:
The notations "SM" (l.211) and "\lambda" (Eq. 10) are used to describe the price of electricity. Please use a consistent notation throughout the paper.

Answer:
Thank you for pointing it out, the notation $SM_t$ is now used throughout the paper.

9.4:
"Equation 8: consider introducing a specific symbol for the RMSE instead of using the abbreviation."

Answer:
The notation of RMSE was changed to $\varepsilon_{RMS}$ and NRSME to $\varepsilon_{NRMS}$.

Detailed comment 10:

10.1:
Please describe in the abstract that the study was conducted for participation on the day-ahead market and for Denmark.
Please include in the abstract the assumption of perfect forecast.

Answer:
The additional information was added.

10.3:
"Sizing of Hybrid Power Plants (HPPs), which include wind power plants and battery energy systems, is essential to capture tradeoffs among various technology mixes": please be more specific.

Answer:
The tradeoff mentioned is an economic tradeoff, that could lead to over- or under-sizing a HPP.

Changes in paper:
The reformulated first sentence of the abstract.

10.4:
l. 4 "model the operation of a battery when participating in any market": please be more specific about the market mentioned here.

Answer 10.4:
The term was changed to electricity market. Later in the abstract, we mention that the study focuses on the day-ahead market.

10.5:
l. 5: "Traditional EMS" : what does "Traditional" mean here? Consider rephrasing.

Answer:
Based on context, we have Replaced "Traditional EMS" with either High-fidelity EMS or LF EMS.

- Minor comments

All minor comments have been addressed, and changes have been made accordingly.

Concerning the comment:
"Be aware that Wind Energy Science guidelines state that grey-literature may only be cited if there are no alternatives. The international hybrid power plant conference is grey literature, due to its lack of peer-review: "Das, K., Hansen, A. D., Koivisto, M., and Sørensen, P. E.: Enhanced features of wind-based hybrid power plants, Proceedings of the 4th International Hybrid Power Systems Workshop, 2019."
"

We have contacted the Workshop organizers and received the following reply:
"
That's partly correct, we only review the abstracts, in the short time between paper deadline and the workshop (about 4-6 weeks) we cannot completely review 180 papers. But those papers which are published in the IEEE Explorer should not be considered gray-literature as IEEE is running them

though their quality check... also, in the last few years we published the proceedings in a digital data base, see https://digital-library.theiet.org/content/conferences/cp847, so if you mentioned the ISBN Number in the reference, it should qualify as a reference.

However, we only started with the digital data base in 2021, but all older proceedings also have an ISBN number and the proceedings have been submitted to a number of University library in Europe, so papers could be found by interesting parties. The relevant reference for the 2019 workshop is:

Proceedings 18th International Workshop on Large-Scale Integration of Wind Power into Power Systems as well as on Transmission Networks for Offshore Wind Plants
Dublin, Ireland, 15-16 October 2019
ISBN: 978-3-9820080-5-9
"

---

## Author Comment (AC2)

**Reply on RC2 – PDF Format**

- The abstract indicates that the EMS "is introduced to model the operation of a battery," which sounds like a narrower scope than the wind-battery system described in the paper.
  Answer:
  Changed wording in abstract to expand scope to wind + battery operation.

- The RMS errors noted in the abstract are not especially meaningful when given as numerical values (to the reader who doesn't yet know what are the scales of the metrics being evaluated). It would be more useful to give these as a percentage or to add the relevant context.
  Answer:
  Removed RMSE of hourly data and added Normalized RMSE (NRMSE) of yearly revenues in percentages. Added ranges for RMSE of profitability index.

- Table 1 certainly motivates that using a high-fidelity EMS can be computationally expensive, but without direct comparisons in your own results it is difficult to assess the relative value of the SM approach.
  Answer:
  We have now removed this Table and replaced it with text in l. 76-102 where the computational burden of the high-fidelity EMS is detailed. The comparison between the HF EMS and the surrogate is highlighted in Section 5.4 and Section 6 l. 549-551.

- Many abbreviations are defined multiple times (e.g., SM on both lines 113 and 122); please check that all are defined only the first time they are used. (I see PI and HPP re-defined as late as p. 18.)
  Answer:
  Changes were carried across the paper.

- Dispatch intervals are given as both 15 min (line 126) and 5 min (line 135). I assume from other parts that 5 min is a typo but please clarify if not.
  Answer:
  Indeed, it was a typo. Thank you for pointing it out.

- There are several 1-sentence paragraphs that interrupt the flow. For example the sentence on line 139 (introducing Figure 1) could easily be combined with the paragraph starting on line 140 (and similar in subsequent instanced).
  Answer:
  All 1-sentence paragraphs are now combined with their corresponding paragraphs.

- Table 3 clearly indicates many variables and constraints but it would be useful to tie these more explicitly to the computational burden noted as a goal for the proposed surrogate model. Is this table related to the 47-min computation noted on line 148?
  Answer:
  These indeed refer to the 47-minute computation time. In the text, we explain that each iteration of the MILP and MIQP problem is solved quickly, in less than 0.15 seconds (l. 223-224). However, many of these optimization problems must be solved sequentially, leading to the 47-minute computation time for one year of input data. The text was slightly modified to make it more explicit that we are referring to the HF EMS on which the surrogate is based.

- (1)-(3) are ratios, not equations as stated. Either give them (short) variable names or omit the equation treatment (you use the ratios as is later in the paper, e.g., line 205)
  Answer:
  Thank you for noting it. The ratios are included in the text, and the equations were removed.

- I struggled at times to understand which surrogates were being described and analyzed. It would be very helpful to give the four surrogates in Table 5 names (e.g., S1, S2, etc.) and then use these names consistently throughout the rest of the paper (e.g., "…surrogate S1…")
  Answer:
  Thank you for the suggestion. The surrogates have been named S1-S4, and the text, figures, and tables were modified accordingly.

- Captions in general are quite short and could be more descriptive. As one example, it would be easier to understand Figure 2 if the 2 sentences on lines 210-211 explaining the nomenclature were in the caption instead of the body text.
  Answer:
  The captions have been changed so that most figures are more descriptive. Additionally, all metrics and variables are now explained before each figure.

- Also in Table 5 I assume that "FFN" is a typo and it should be "FNN"; otherwise, please explain.
  Answer:
  Indeed, it is a typo.

- It is not clear what would be the desired level of truncation for the principle component matrices Z (line 220); how was this desired level selected?
  Answer:
  Thank you for pointing that out, I apologize for the oversight. Additional text is now added to explain that the truncation level is such that we have an explained variance of 99%: l. 278-280.

- Line 257 is missing the word "Appendix"
  Answer:
  Word added.

- Please clarify if the y terms in (8) are for the normalized or actual values in the time series. Line 266 suggests normalized but 269 and 270 discuss true and predicted data without the normalization qualifier. This will also impact the quality of the results as measured by RMSE (i.e., relative to a scale of 0-1 or a much wider scale from the original data).
  Answer:
  Note that Eq. (8) is now Eq. (5).
  The y-terms refer to the normalized values. The variables within the RMSE' equation (5) are now modified for clarity. Additionally, the figures now show explicitly when the normalized variables are used.

- Why is the text below (9) only appearing in a subset of the page width?
  Answer:
  It is now integrated into a paragraph.

- Line 300 explains PI in words, but the equation doesn't appear until approximately a half page later; could be more streamlined to just have the equation in the paragraph where it is introduced.

Answer:

Thank you for the suggestion. We have included the equation for the PI in the paragraph.

- I recommend avoiding use of longer, non-standard words as "symbols" in equations, e.g., "Profit_y" in CF_y on line 319, as this makes the equations harder to read. (I understand that CAPEX and OPEX are often used as such in equations in wind energy related publications.)
Answer:

We have changed some variables according to the comment.

- Line 341: I assume it should be ($y_b(i_b)$) (subscripts) as in the equation.
Answer:

Indeed, thank you for pointing it out.

- Lines 354-355: these variables have been previously defined
Answer:

This is now corrected.

- Lines 362-363: I don't know what "This tool is based on re-analyzes…" means. Re-analyzed?
Answer:

There was a typo; it is now changed to "meteorological reanalysis data."

- Please ensure adequate font size in all figures (especially Figure 4)
- Typo in Figure 4b caption (should be Normalized prices)
Answer: Figure 4 is now removed.

- Section 4.1.3: are 250 or 200 HPP configurations studied? The end of p. 16 says both.
Answer:

The paragraph of this section is now modified for clarity: there are a total of 250 HPP configurations used, 200 for training and 50 for validation. All configurations are unique.

- Figure 5(b) is an interesting way to visualize the different probability distributions but needs explanation since it is non-standard. Also, the y-axis needs units (MW?)
Answer:

An additional explanation is now included to explain the plot: l. 413-426. As the y-axis has normalized wind power generation, it is unitless.

- Please ensure that all results in Section 5 (figures, tables, and text) are clear about which surrogate model has been used (referring to the suggestion to give them names in Table 5)
Answer:

The results in this section now refer to the best-performing surrogate, model S4. This has been clarified in the text. The comparison of the performance of all surrogate models is now moved to Appendix A.

- For Figure 6 and 7 (and related discussion), I refer back to my question about whether the RMSE is based on the normalized data to help the reader evaluate the quality of the method.
Answer:

Note that Figures 6 and 7 are now Figures A1 and 5.

For both figures we use normalized data. For Figure 5, we now explicitly mention it in the text and on the figure by using the normalized variables as the x-axis labels. For figure A1 we mention in the text leading up to the figure that we use the normalized data.

- For Figure 8, instead of noting "MegaWatts" in the caption it would be better to include "(MW)" in each of the y-axis labels
  Answer:
  This has changed.

- On line 435 and related discussion you mention the "density" of the data points but the colorbar on Figure 9(a) has units of "count". I understand that these are related but more precise language would be more clear.
  Answer:
  This has been clarified in the text: l. 468.

- Line 477: please name the surrogate used instead of "the selected surrogate" here, as well
  Answer:
  This has been modified as suggested.

- Line 515: "…all HPP configurations are not profitable…" has a different meaning than "…not all HPP configurations are profitable." I think you mean the latter and should therefore revise accordingly.
  Answer:
  This has been modified according to the suggestion. Thank you.

- Typos and grammar to change:
  -line 17: "wind power plants are" (should be plural)
  -line 20 appears to be missing a space between ".This"
  -line 42: "accurate forecasting can mitigate these penalties" should be proceeded by ; (not a comma) or a standalone sentence
  -Table 1: "Iterations" should be plural
  -line 90: the sentence starting "Two of which" is incomplete
  …and so on. I recommend a close re-reading as part of the revision process to address these and similar errors throughout.
  Answer:
  Several of these mistakes have been modified after a closer re-reading.

- Furthermore, I believe the citation format is not aligned with WES standards (Author, Year) in most cases except where the author's name is part of the sentence (e.g., "Author (year) showed that…").
  Answer:
  This has changed.

---

## Author Comment (AC3)

**Reply on RC1 – PDF Format**

- Detailed Comment 1:

Figure 8 reports the output time series for the high-fidelity model and for the surrogate model. It can be seen on the figure that at t=0 h the charge power is positive and the discharge power is negative. One would expect the discharge power to be zero. Furthermore, the power output of the HPP is negative between t=2 and 4 h, assuming that the HPP is able to charge from the grid. These two points raise questions on the ability of the surrogate model to represent a valid storage model. Please expand the discussion of Figure 8 to include these considerations.

Answer 1:

The purpose of showing the time series in Figure 8 (now Figure 6) is to demonstrate the hourly performance of the surrogate. While the trends and fluctuations within the HPP output are well captured, the same cannot be said for the other time series. This highlights the ability of the surrogate to represent the HPP operation as a whole, but its inability to describe the hourly operation of each technology individually with high accuracy. These issues arise from the nature of the regressor; the FNN cannot inherently capture the physical constraints that multiple equations would typically enforce. Specifically, the FNN lacks explicit equations to govern its outputs. However, for this study, focusing primarily on the power output of the HPP is sufficient, as this is the only variable required in revenue calculation and subsequent profitability index evaluation.

Changes in paper:

Additional discussion is added in Section 6 to elaborate on that. l. 523-531

"However, challenges persist, particularly in estimating battery charge and discharge profiles. As depicted in Fig. 5 and 6, this difficulty arises from the high-fidelity model's abrupt power output fluctuations and the intrinsic non-linearity of these variables. In particular, Fig. 6(c) and (d) highlight the surrogate's limitations in accurately and consistently representing the battery's hourly operation. In several time steps, the model shows the battery charging and discharging simultaneously, and at times, the charge and discharge power take on negative values, which is physically unrealistic. These issues stem from the nature of the regressor; the FNN cannot inherently capture the physical constraints that would typically be enforced by multiple equations. Specifically, the FNN lacks explicit equations to govern its outputs. However, for the purposes of this study, focusing primarily on the power output of the HPP is sufficient, as this is the only variable required in revenue calculation and subsequent PI evaluation."

- Detailed Comment 2:

The story line of the paper focuses on the field of system integration of renewable energy and hybrid power plants. In order to fit better in the scope of Wind Energy Science, consider highlighting the relevance of this study for the field of wind energy in the introduction, discussion and conclusion of the manuscript.

Answer:

Additional text was added at the start of the introduction and discussion and at the end of the conclusion to highlight the study's relevance.

- Detailed Comment 3: Motivation and research question:

3.1:

l. 37: " The importance of a realistic EMS …" : it is unclear from the literature review what is the difference between a high- and low-fidelity model for the EMS. The manuscript provides an

overview of different high-fidelity models used in the field but does not describe the low-fidelity ones. The following reference may be of interest in this context:
Stanley, A. P. J., & King, J. (2022). Optimizing the physical design and layout of a resilient wind, solar, and storage hybrid power plant. Applied Energy, 317, 119139. https://doi.org/10.1016/j.apenergy.2022.119139

Answer:
To explain the differences between the High-Fidelity (HF) and LF (EMS), additional literature was added within the introduction (l. 59-77), and a new table was added: Table 1. Comparison of HF and LF EMS Models.

"EMS models vary in complexity and computational demand, and for this article, we categorize them into high-fidelity (HF) and low-fidelity (LF) models. High-fidelity EMS models provide detailed and accurate representations of HPPs, capturing intricate system dynamics, precise component behaviors, and sophisticated market interactions. These models incorporate forecasting and real-time data, comprehensive operational constraints, and optimize bidding strategies to maximize profits in electricity markets Taha et al., 2018; Zhu et al., 2024; Ochoa et al., 2022; Han and Hug, 2020; Li and Qiu, 2016; Abdeltawab and Mohamed, 2015; Yang et al., 2018; Huang et al., 2021; Das et al., 2020. While HF EMS models offer high accuracy in estimating operational performance and financial outcomes, they require significant computational resources and time. For instance, Huand et al. (2021) develops a stochastic optimization-based HF EMS where solving the dispatch for one week of operation takes between 329 and 2,991 seconds, depending on the chosen optimization algorithm among five compared. Similarly, Li and Qiu (2016) present a deterministic HF EMS model that requires using a monthly time step to reduce simulation costs.

In contrast, LF EMS models simplify the representation of HPP operations by using aggregated system models, basic forecasting methods (or none), and simplified market participation strategies (An et al., 2020; Luo et al., 2015; Cai et al., 2016; Zhang et al., 2018, 2017). They reduce computational demand by neglecting detailed component behaviors and operational constraints. Although they enable rapid simulations and are easier to integrate into optimization frameworks, their oversimplifications can lead to inaccuracies in revenue and cost estimations, potentially resulting in sub-optimal or erroneous sizing decisions. Table 1 provides an overview of the distinctions between HF and LF EMS models. Notably, there is no universally accepted standard for defining these classifications, so the table serves as a guideline based on key characteristics relevant to this study."

**Table 1.** Comparison of HF and LF EMS Models

| Aspect | HF EMS | LF EMS |
| --- | --- | --- |
| Component modeling | Physical modeling per component considering the electrical, mechanical, and/or thermal behaviors. Includes battery degradation model | Simplified or aggregated model, linear approximations or average values used for component performance |
| Market Participation Modeling | Complex bidding strategies in various electricity markets (day-ahead, intraday, balancing), includes market rules and regulations (i.e., dispatch and settlement intervals), and/or grid compliance requirements | Usually focuses on one market, ignoring opportunities or penalties from other markets |
| Input data and Forecast | Varying resolution (from sub-hourly to hourly). Possible combination of forecast and real-time data | Uses hourly resolution. Deterministic forecasts or perfect forecast |
| Optimization problem formulation | Complex formulations like: mixed-integer linear programming (MILP), nonlinear programming (NLP), or stochastic optimization | None e.g. rule-based, or linear programming |
| Computational Demand | High computational demand due to complex modeling and optimization, leading to long simulation times | Low computational demand, allowing for fast simulations and scalability |
| Accuracy | High | Low |

3.2:
Table 1 reports that 1000 "iterations [are] required to find [a] refined solution". Please put this number in context with the literature. Is it characteristic of HPP sizing problems to require a large number of iterations to converge?

Answer:
The short answer is that from previous work that authors are familiar with (Leon et al., 2024), we need to evaluate hundreds of different sizes to reach an optimal sizing.

Changes in Paper:
The previous Table 1 is now removed and replaced with another table, Table 1. Comparison of HF and LF EMS Models. Instead, we provide more details on the sizing problem in the introduction l. 78-100.

"The trade-off between computational efficiency and model accuracy presents a significant challenge for the optimal sizing of HPPs. A sizing optimization of an HPP involves maximizing a financial metric by varying the wind power plant rating, battery energy, and power ratings. Calculating that financial metric requires solving an EMS model for each potential HPP configuration. Consequently, HF EMS models offer precise assessments; however, relying solely on them is impractical due to their substantial computational demands. Conversely, using LF EMS models reduces computational time but risks compromising the financial viability of the project due to inaccurate assessments.
To illustrate the computational burden of an HF EMS model, we evaluate the state-of-the-art EMS developed by Zhu et al. (2022) This model requires 1,250 minutes to solve for 25 years of operation (the assumed lifetime) of a given HPP using a single-node High Performance Computing (HPC) cluster, Sophia (DTU HPC Cluster, 2019), which has 32 physical cores (2 × sixteen-core AMD EPYC

7351) and 128 GB of RAM (4 GB per core, DDR4@2666 MHz). Therefore, even if we need to evaluate only a few sizings for the optimizer to converge, we require a substantial amount of time to reach a solution. For example, evaluating 10 sizings takes 12,500 minutes, or approximately 208 hours. Additionally, in previous work familiar to the authors Leon et al. 2024, a sizing optimization can take up to several hundred iterations to approach optimality. In that study, the authors use a low-fidelity EMS model to evaluate the operation of an HPP over its lifetime in a matter of 15 seconds. The comparison of the optimization time is based on the same computational resources.

Given these computational benefits, HPP sizing optimization often relies on LF EMS models. For example, Leon et al. 2024 propose a methodology for sizing HPPs as a nested optimization problem, using two LF EMS models: a short-term EMS formulated as linear programming and a long-term rule-based EMS. The short-term EMS provides a baseline for daily optimal operations, while the long-term EMS modifies these operations to account for degradation effects and forecast inaccuracies over the plant's lifetime. Similarly, in a study aimed at optimizing the design and layout of a hybrid wind-solar-storage plant, Stanley and King (2022) employs a simple battery dispatch model, where the battery is only discharged to meet minimum power requirements. While using LF EMS models may result in reduced accuracy in revenue estimation, they are widely adopted in HPP sizing due to computational efficiency. Indeed, several review studies underscore the prevalence of LF EMS models in sizing methodologies (Roy et al., 2022; Lian et al., 2019; Thirunavukkarasu et al., 2023)

3.3:
l. 59: "HPP sizing optimization often relies on a simplified EMS representation": the introduction of the paper does not describe the problems associated with using a simplified or low-fidelity EMS model. As such, it is unclear why one would prefer a high-fidelity model in a context where computational cost is a problem. Please describe explicitly and quantitatively why a high-fidelity model is superior to a low-fidelity one.

Answer:
Although no studies directly compare HF and LF EMS models, it is evident that LF EMS models sacrifice accuracy due to simplifications in areas such as component modeling, market bidding strategies, and operational constraints. These simplifications can result in inaccurate power schedules, leading to revenue projections that may misrepresent the business case for a given HPP sizing. Quantifying the extent of deviation between low- and high-fidelity EMS models is beyond the scope of this paper; however, we are currently researching this topic.

Nonetheless, Zhu et al. 2024 examined the accuracy of total profits across three HF EMS models, demonstrating that even among HF models, certain simplifications commonly found in LF models— such as relying on deterministic forecasts—can lead to revenue discrepancies of up to 7.6% when compared to the best-performing model (refer to Table 3 of the paper).

Although this provides some insight into the potential impact of LF EMS simplifications, a comprehensive study comparing HF and LF EMS models for HPPs is currently being conducted by colleagues in our research group.

Changes in paper:
A paragraph (l. 101-117) was added to explain the current state of the research on comparing HF and LF models.

"It is challenging to quantify the accuracy loss when using LF EMS instead of HF models. Research studies often test EMS models on varied configurations of HPPs, and only a few conduct direct comparative analyses within the same setup, primarily focusing on high-fidelity EMS models. For

instance, Ochoa et al (2022) compare deep reinforcement learning with both stochastic optimization and robust optimization for photovoltaic-battery HPPs using U.S. market data, finding that deep reinforcement learning offers superior economic performance and significantly reduces computational time compared to the other two studied methods. Similarly, Han and Hug (2020) report that, in a one-year simulation using Nord Pool data, the distributionally robust optimization model achieves higher revenues than deterministic forecasting approaches. Zhu et al. (2024) further explore this by comparing EMS models that utilize distributionally robust optimization with those based on deterministic optimization and stochastic optimization for wind-battery hybrid plants in Nordic day-ahead markets, taking imbalance settlements into account. By adjusting the parameters of the distributionally robust optimization model, they demonstrate that the economic performance ranks highest for this approach, followed by risk-neutral stochastic optimization, and finally deterministic optimization. This approach enables more resilient offering strategies, especially in markets with high penalties for energy imbalances.

Additionally, Zhu et al. (2024) examine the accuracy of total profits across three HF EMS models and show that even within these models, certain simplifications commonly found in low-fidelity models—such as the use of deterministic forecasts—can lead to revenue discrepancies of up to 7.6% compared to the best-performing model (refer to Table 3 of the referenced paper). Given these considerations, this paper primarily focuses on reducing the computational demand of high-fidelity EMS models."

3.4:
The list of major contributions is a good addition to the introduction. Consider writing explicitly the research question for the study.

Answer:
The research question was added before the list of contributions. l.162-164.

"Building upon the EMS model developed by Zhu et al. (2022), which is detailed in the following section, this paper seeks to answer the question: How can we enable the sizing evaluation of utility-scale HPPs based on an accurate and computationally efficient EMS model?"

3.5:
l. 100-101: "Integration of the developed surrogate within a framework to evaluate the profitability of an HPP sizing with high accuracy.": this statement is convoluted. Consider combining it with the previous one, e.g.: "Assessment of the surrogate model's ability to calculate … time series and profitability of the HPP …"

Answer:
As the three points are distinct, we would like to keep them separate. The wording was modified to make it more straightforward.

Changes in Papers:
The second and third points were reformulated (l.167-170).
"
- Demonstration of the surrogate's generalizability in different geographical locations within the same electricity market region.
- Integration of the developed surrogate within a sizing evaluation framework to accurately assess the profitability of various HPP configurations.
"

- Detailed Comment 4: Methodology

4.1:

Two of which use a multivariate linear regression to establish a baseline and two others are based on Neural networks (NNs)": please put in context the choice of surrogate models in the introduction. What are the strengths and weaknesses of these models? Have they been applied to models similar to HPP dispatch strategies? Can one expect them to perform well for this type of problem? Are there other surrogate models one could consider for HPP dispatch strategies?

Answer:

In the revised introduction, we have discussed data-driven surrogate models, showcasing successful applications of both regression models—ranging from linear to more complex Neural Networks (NNs). These models are often applied to solve problems involving partial differential equations where a large number of parameterized instances must be evaluated. In such cases, thousands of degrees of freedom are typically required to achieve accurate solutions, leading to significant computational demands, especially in scenarios requiring real-time simulations.

This challenge is similar to our current problem, where the high-fidelity EMS model must be solved across hundreds of HPP sizing configurations. For each sizing, the EMS model operates at an order of magnitude involving hundreds of thousands of degrees of freedom, creating a substantial computational burden. Given that the use of surrogate models for EMS in grid-connected HPPs is largely unexplored in the literature, we examined a range of potential surrogate models. Our exploration includes simple approaches, such as linear regression, and more sophisticated models, like feedforward neural networks (FNNs), initially chosen based on their adaptability to our specific problem.

The linear regression model serves a dual purpose: firstly, to assess if a linear approximation can capture the essential dynamics of our problem, and secondly, to provide a baseline against which we can measure the improvement in accuracy and computational cost when using more complex surrogate models.

Changes in paper:
Additional text was added: l.118-147

"To address the computational challenges associated with implementing a realistic EMS for HPP sizing while maintaining high accuracy, a promising approach involves using data-driven surrogate-based modeling. This technique demonstrates potential in tackling computationally intensive problems across various domains (Zhang et al., 2021; Lin et al., 2023; Pang et al., 2023). These Reduced-Order Models (ROM) aim to replace high-dimensional, resource-intensive problems with models that are significantly faster to simulate while accurately representing the original solution behavior. In particular, Hesthaven et al. (2022) reviews the development of surrogates for time-dependent problems, including those with nonlinear dynamics, which are of interest in our work. In this context, data-driven surrogate models stand out as promising solutions, thanks to major advancements in machine learning methods.

These models often follow an offline-online paradigm. During the offline phase, a reduced basis is extracted from a collection of high-fidelity solutions; this reduced basis is then used to train the surrogate model by optimizing weights or coefficients that capture the system dynamics. Although this step is computationally intensive, it only needs to be performed once. In the online phase, the surrogate model uses the precomputed weights to compute new outputs almost instantly, with minimal computational cost. This paradigm enables the surrogate model not only to learn the mapping from inputs to outputs but also to understand underlying patterns within the input data, leading to faster and more accurate simulations.

Numerous successful implementations of data-driven surrogate models exist in the literature. For instance, Hesthaven and Ubbiali (2018) develops an ROM using Proper Orthogonal Decomposition (POD) to extract a reduced basis from high-fidelity solutions and employs multi-layer perceptron neural networks to approximate the coefficients of the reduced model, although time-dependency is not considered. Similarly, Guo and Hesthaven (2019) uses a POD projection and maps the time and parameter values onto the reduced basis using tensor products of two Gaussian processes—one for time and one for parameters. Hess et al. (2023) utilizes a data-driven ROM approach to efficiently compute the Rayleigh–Bénard cavity problem, integrating POD, dynamic mode decomposition, and manifold interpolation for a robust and computationally efficient model. Departing from POD, Bhatt et al. (2023) employs deep auto-encoder networks to compress high-fidelity snapshots before using these in forecasting models—specifically, long short-term memory and temporal convolutional networks for time-series forecasts, and convolutional neural networks for spatial feature extraction—significantly reducing computational costs in both the offline and online stages.

Most ROMs have been applied to problems described by partial differential equations with sharp gradients. In contrast, our aim is to apply similar techniques to high-fidelity EMS models for HPPs. Although surrogate modeling has advanced across multiple fields, a gap remains in developing models tailored to EMS for utility-scale HPPs, particularly those that incorporate detailed operational strategies for market participation. This gap exists not only due to the scarcity of existing applications of surrogate models for EMS in HPPs but also because of the complexity involved in designing an accurate surrogate model based on a multitude of input and output time series. Additionally, integrating the surrogate model within a sizing evaluation framework adds another layer of complexity."

4.2:
In the first part of section 5, the four surrogate models are compared. However, only one model is used for the rest of the results section. This leads to a lengthy result section, where the high impact results are more difficult to identify. Consider either (i) focusing on the best surrogate model in the entirety of section 5 and moving the comparison between linear and NN models in an appendix; or (ii) compare all four models for all relevant metrics. This second approach would help the reader have insight on the trade-off between accuracy and computational time.

Answer:
Thank you for the suggestion. We decided to go with the first option, and changes were made in Section 5 accordingly.

Changes in paper:
Figure 6 is now Figure A1, and Table 10 is now Table A1.
Moved previously numbered Figure 6 (now Figure A1) and Table 10 (now Table A1) to Appendix A: Surrogate Models Comparison and the corresponding text.
Only figures related to the best-performing surrogate models (S4) were kept in Section 5.

4.3:
The surrogate models are compared to each other, with the high-fidelity EMS as a reference. However, the results would be significantly strengthened if the comparison were to include a low-fidelity model. For example, if the RMSE for the low-fidelity EMS was 10 times higher than the linear and NN models (Figure 6), this would provide an excellent motivation for the use of a surrogate.

Answer:
Thank you for the suggestion. While I agree that including a comparison with a low-fidelity EMS would provide additional context and further strengthen the results, the focus of this study is

specifically on the use of surrogate models as a computationally efficient alternative to high-fidelity EMS. As Zhu et al. (2024) demonstrated, even minor simplifications typical of low-fidelity EMS can lead to significant revenue discrepancies, with observed differences reaching up to 7.6% compared to the most accurate high-fidelity model. This suggests that the discrepancy would be even greater with a low-fidelity EMS.

Additionally, our colleagues' ongoing research explicitly addresses the comparison between high- and low-fidelity EMS models. While the insights from that work would be valuable here, incorporating it is beyond the current scope of this paper. Instead, this study focuses on evaluating the accuracy and computational benefits of surrogate models relative to high-fidelity EMS, as this comparison more directly aligns with the primary goals of the research.

4.4:
l. 208: "applied for the 2nd and 4th surrogate models": consider introducing a name for each surrogate, instead of referring to their number in Table 5. The names should match the labels of the figures in the results section.

Answer:
The names of the surrogates have been changed to S1-S4, please refer to Table 5 for the definition of each model. Other changes in the text and figures have been applied.

4.5:
l. 407: "as the linear model cannot capture the inherent non-linearities of the high-fidelity model.": why has a linear model been chosen? This statement suggests that the choice of methodology is not appropriate for the study.

Answer:
Thank you for pointing that out. I will revise the statement to clarify this aspect.
In addition to the reasons mentioned earlier for selecting the linear regression model, we recognize that the EMS exhibits some non-linear behaviors. However, we also suspect these non-linearities are relatively mild, as most of the system's constraints are linear. Therefore, we included the linear regression model to evaluate its effectiveness in approximating the high-fidelity model. This approach allows us to establish a baseline and assess the extent to which a simple model can capture the EMS's behavior before moving on to more complex surrogate models.

Changes in paper:
Sentence added in l.428-430

"This result is expected to a certain degree: while the linear model may not fully capture the non-linear dynamics of the high-fidelity model, we selected it to assess the extent to which a simpler model can approximate the EMS, given that many of the HF EMS model's constraints are linear."

- Detailed Comment 5: Literature review
5.1:
On the topic of energy markets and subsidies for renewable energy, consider referring to the following report: European University Institute: Robert Schuman Centre for Advanced Studies, Kitzing, L., Held, A., Gephart, M., Wagner, F., Anatolitis, V., & Klessmann, C. (2024). Contracts-for-difference to support renewable energy technologies : considerations for design and implementation, European University Institute. https://data.europa.eu/doi/10.2870/379508

Answer:
Thank you for the recommendation. The paper is now cited in the introduction, and additional context was added since the writing of this report, e.g., the Agreement of May 2024. l.20-24

"As noted by Kitzing et al. (2024), contracts-for-differences have become increasingly significant in the European market, particularly following an agreement reached in May 2024 between the European Commission, the European Parliament, and the Council (Council of the European Union, 2024). This agreement mandates that the support for renewable technologies is provided through two-sided contracts-for-differences or equivalent schemes, applying to both new investments and repowering."

5.2:
The literature review would be strengthened by citing literature related to machine learning and data-driven methods. An overview of methods for modelling time-series would be a good addition to the paper.

Answer:
Additional literature on these topics has been added. l.118-147

Please refer to answer 4.1 for the additional text.

5.3:
Consider including a short description of the advances done on hybrid power systems (e.g. for micro-grids). This would help contextualize better the paper since the problem of storage sizing and dispatch strategy has been addressed in this field before, even though not in relation to electricity markets.

Changes in paper:
Additional context was added in l.53-58.

"In the field of hybrid renewable energy systems, particularly in microgrids, the dispatch problem has been extensively studied, as evidenced by numerous review articles on the topic (Barbosa et al., 2024; Shivarama Krishna and Sathish Kumar, 2015; Fathima and Palanisamy, 2015). However, the primary purpose of microgrids is to manage or follow load profiles within a network, whereas HPPs operate as distinct generation facilities with a connection to the power grid. This unique connection emphasizes the role of HPPs in active power generation, rather than load management alone, which calls for a distinct EMS model such as the ones developed by Toubeau et al. (2021) and Ding et al. (2016)."

5.4:
The literature review could be complemented by citing articles related to the field of bidding strategies. For example, the works by Pierre Pinson or Kenneth Bruninx may be of interest:
Ding, H., Pinson, P., Hu, Z., & Song, Y. (2016). Integrated Bidding and Operating Strategies for Wind-Storage Systems. IEEE Transactions on Sustainable Energy, 7(1), 163–172. https://doi.org/10.1109/TSTE.2015.2472576
Toubeau, J.-F., Bottieau, J., De Greve, Z., Vallee, F., & Bruninx, K. (2021). Data-Driven Scheduling of Energy Storage in Day-Ahead Energy and Reserve Markets With Probabilistic Guarantees on Real-Time Delivery. IEEE Transactions on Power Systems, 36(4), 2815–2828. https://doi.org/10.1109/TPWRS.2020.3046710

Answer:

Thank you for the suggestion. These papers have been cited.

Changes in paper:
Please refer to answer 5.3 for the additional text containing these citations.

- Detailed Comment 6: Structure of the paper

6.1:
The last sentence of the first paragraph states that BESS are valuable to establish robust business cases. Then, the second paragraph discuss the definition of HPPs. In this case, the link between the two paragraph is not clear. Instead, consider introducing HPP in the first paragraph, and narrowing the focus of the study on HPP with storage systems.

Answer:
Both paragraphs were modified according to the comment: l. 27-36

Changes in paper:

"Hybrid Power Plants (HPPs) that include storage technology are becoming valuable to establish robust business cases. Despite the absence of a universal definition for HPPs — highlighted by varying interpretations in the literature (Dykes et al., 2020; Long et al., 2022; Paska et al., 2009) — for the purposes of this research, we define renewable-based HPPs as power plants that combine several generation technologies, including wind turbines and possibly energy storage, to produce electricity and other energy vectors. They operate from a single geographical location, with all generated power being transmitted to the electrical grid via a unified point of grid connection. In the presence of a BESS, HPPs can use strategies such as market arbitrage, which involves buying and storing electricity when market prices are low and selling it when prices are high. Additionally, the integration of energy storage is crucial for smoothing power supply fluctuations, mitigating power curtailment, enabling HPPs to offer system services, and reducing grid congestion (Das et al., 2019)."

6.2:
l. 68- 71: "To evaluate the value of HPP, …": this paragraph discussing performance metrics for HPPs does not seem relevant in the introduction. Consider moving it in a later section describing the profitability index.

Answer:
Agreed. This sentence was integrated in Section 3 (l. 336-347), where financial metrics are discussed.

Changes in paper:

"To assess the business case of an HPP, we can use financial metrics like Internal Rate of Return (IRR) and Net Present Value (NPV). IRR calculates the HPP's annual investment return, while NPV assesses its profitability in today's value. However, when an HPP isn't profitable, resulting in a negative NPV, the IRR becomes undetermined. A more meaningful measure is the Profitability Index (PI), calculated as $NPV/CAPEX$. The PI indicates how many dollars of present value benefit
are generated per dollar of investment, offering a more intuitive understanding of the investment's profitability. This metric allows for a direct comparison of the relative profitability of each project, regardless of their absolute size. Additionally, when resources are limited, NPV/CAPEX can aid in prioritizing projects. Projects with higher PIs can be prioritized as they promise greater returns per unit of investment. A PI greater than 1 signifies that the NPV of future cash flows exceeds the initial investment. Note that traditionally, for power plants using one type of generation technology, the

Levelized Cost of Energy (LCoE) is used to evaluate profitability. However, to assess the various potential revenue streams, metrics such as NPV (Dykes345

et al., 2020) or the NPV/CAPEX are more relevant. This is because storage inherently increases costs and thus the LCoE, even though it has the potential to substantially increase revenue or profit."

**6.3**
l. 118-125: "In electricity trading …": the description of the spot and balancing market does not help describing the methodology of the study. Consider moving this paragraph to the introduction.

Answer:
The paragraph was shortened and integrated in the introduction: l. 41-52.

Changes in paper:
"The Spot Market (SM) is currently the most lucrative market where power is traded for immediate delivery. Within the SM, operators can participate in the day-ahead and hour-ahead markets. Day-ahead bidding establishes hourly prices for the following day, while hour-ahead
bidding allows for adjustments based on updated generation forecasts and cleared SM prices. Since electricity markets require power producers to bid in advance on the amount of electricity they will generate, these bids rely on forecasts of renewable energy generation and market prices. If actual generation deviates from scheduled bids, financial penalties are incurred. Additionally, the balancing market (BM) offers HPPs another revenue source, operating alongside the SM to handle discrepancies between forecasted and actual demand and supply. The BM allows transmission system operators to adjust for differences arising from SM bids that are based on earlier forecasts and the real-time conditions closer to delivery. This market acts in near real-time, addressing deviations from scheduled generation and imposing penalties as necessary. Consequently, an EMS aims to maximize profits by strategically storing and selling electricity in both the SM and BM, while also minimizing imbalance costs due to deviations from scheduled energy bids."

**6.4**
Consider restructuring section 2 and 3 into one section describing the metrics relevant to HPPs (including the description of the relevant time series, the EMS and the profitability index) and one section describing the surrogate models.

Answer:
After discussing this with most co-authors, the majority wished to keep the current structure.

**6.5**
l. 185-186 "Details on the normalization process appear later on" and l. 189 "The specific use of this method is detailed in this section": instead of referring the reader to a later part of the paper, consider restructuring the subsection.
Consider shortening the description of the normalization steps (l.194-205) and instead state that scaling is used for all time series.

Answer:
The subsection was slightly restructured, and the description was shortened as suggested.

**6.6**
l. 213-218: the description of the shapes of the matrices does not seem relevant for the study. Consider removing the associated sentences or moving them to an appendix.

Answer

These descriptions were removed.

6.7
Please restructure and shorten section 2.2.3. The text mixes general statements about training neural networks, mentions of the steps of the methodology (l. 235 and l. 255 "the best-performing model … is selected") and descriptions of the training methodology. This makes the subsection difficult to follow.

Answer:
It seems there is some confusion. The selection of the best-performing model is part of the training process. This does not refer to selecting the best-performing model among the four types of surrogates, S1-S4; instead, this refers to one surrogate type, e.g., S3 or S4. In the tunning process, several hundred models are evaluated for a given type of surrogate (S3 or S4). These hundreds of models differ in the choice of hyperparameters. Among these, the best performing model is selected and further trained "till convergence." The training method described needs to be applied individually for surrogate types S3 and S4.
A workflow example for model S3 would be as follows:
Apply normalization to inputs and output --> define FNN architecture and hyperparameter search space --> Proceed with tuning --> Result: hundreds of FNN trained --> Get the most accurate FNN based on the defined loss function (MSE) --> Further train that model till convergence --> Result: model S3.

The section was modified and shortened to make it more straightforward.

Please let us know if we have misunderstood your comment.

6.8
Consider moving the description of the cost model to an appendix (l. 322-344)."

Answer:
The description of the cost model has now been moved to Appendix C: Cost Model. Moreover, the data related to the cost model has been moved; previously, Table 9, now Table D2, has been moved to Appendix D: Data Supplement.

- Detailed Comment 7: Clarity and conciseness

7.1
l. 25 "power plants that combine several technologies": please precise the type of technology. Consider using the terms "electricity generation and storage technologies".

Answer:
The wording was changed to "combine several generation technologies, including wind turbines, and possibly energy storage": l.30-31

7.2
l. 32-35: "As HPPs transition to market-driven revenue models… throughout the power plant's lifetime": the start of this paragraph is vague. What are the "new possibilities and challenges" mentioned? What does the expression "navigate energy markets" mean in the context of the study? What are the characteristics of a "detailed operational strategies"?

Answer:
A paragraph was added before explaining the meaning of market-driven models, i.e., CFD. The text was further modified to explain the opportunities, challenges, and detailed operational strategies. l. 37-52.

Changes in paper:
Please refer to answer 6.3 for the modified text.

7.3
l. 35 "Energy Management System": please define the term, and highlight the difference between other types of "control" in the context of HPP. Consider stating the difference between EMS and adjacent terms such as bidding or dispatch strategies.

Answer:
A first definition of the EMS is given in the introduction l. 38-41 (please refer to answer 6.3 for the modified text). A more detailed definition is now given in Section 2.1 to clarify which EMS is used in this article. l. 188-197.

Changes in paper
l. 188-197: " However, this paper primarily focuses on the EMS's role in day-ahead SM participation. Additionally, our study considers the Danish market structure with a dispatch interval of 15 minutes. As in real power plants, the SM bidding process (also referred to as SM optimization) communicates with a Real-Time (RT) dispatch optimization. In this framework, the SM optimization provides energy set-points based on weather and market forecast data to the RT dispatch, which, in turn, uses real-time measurement data to derive real-time power values. Real-time measurements allow the calculation of deviations and the application of penalties. The inputs to the SM optimization are time series forecasts of wind power and market prices, while the RT dispatch uses the same input time series updated with real-time measurements for each dispatch interval, as well as the bidding schedule generated from the SM optimization. For clarity, the inputs and outputs of the SM optimization and RT dispatch are listed in Table 2. The combined models, SM optimization and RT dispatch, are referred to as a high-fidelity EMS model in this
paper. "

7.4
l. 95: "Development of a fast and precise surrogate": the term "accurate" seems more pertinent in this context.

Answer:
The term was modified.

7.5
l. 98 "Assessment of the surrogate's ability to predict hourly operational time series": consider using the verb "compute", "calculate", "model" or "estimate" instead of "predict", since the latter implies a focus on future (and unknown) data.

Answer:
Thank you for the suggestion. Similar changes were applied to the paper.

7.6

Please precise what the term "surrogate model" means in the context of the study. By itself, "surrogate" implies a simplified or approximation model, and does not refer to data-driven or machine learning methods specifically.

Answer:
Additional literature on data-driven surrogate models was added to the introduction to give context to the developed surrogate models. Additionally, the paragraph of section 2.2 gives a detailed definition of the term in this study.

Changes in paper:
For the context of surrogates please refer to l. 118-147. For the text please refer to answer 4.1.
For the definition of section refer to l. 231-238:
"In this article, a surrogate model consists of several sub-components: data pre-processing, a regressor, and data post-processing. The pre-processing involves scaling, which ensures that all inputs contribute equally to the model's estimations and supports the surrogate's convergence algorithm. The post-processing is applied in accordance with the pre-processing to interpret the results in their original scale. The regressor is the model tasked with approximating the high-fidelity EMS. Section 2.2.1 details the inputs and outputs for training and evaluating the surrogate models. Section 2.2.2 describes four surrogate models, differing in their data processing and regressor models. Sections 2.2.3 and 2.2.4 cover the training and validation of these models, respectively."

**7.7**
l. 404: "This RMSE provides a holistic measure of the model's accuracy": why is the term "holistic" used here? Consider rephrasing.

Answer:
The sentence was changed to: "Since this RMSE is calculated across all output time series, it provides a broad assessment of the model's accuracy, without specific insights into each individual series." l.317-318.

**7.8**
l. 413: "Table 10 contrasts the time required to execute the workflow for each surrogate model" Consider rephrasing this sentence.

Answer:
The sentence was changed to: "Table A1 compares the time needed to execute the methodology for each surrogate model." l.584.

**7.9**
l. 426: "Figure 8 shows the difference between the surrogate's prediction and the ideal behavior": what does "ideal behavior" mean here? Consider rephrasing.

Answer:
The sentence was changed to: "Figure 6 shows the difference between the surrogate's approximation and the HF EMS' outputs" l. 451.

**7.10**
l. 537 "the synergistic use of SVD and FFN": what does "synergistic" mean in the context of the study? Consider rephrasing.

Answer:

Changed to: "A key innovation of our study is the combined use of SVD and FNN, which represent a novel approach in this field." l. 570-571.

7.11
l. 542: "a mere 25 seconds" and "remarkable accuracy": Please avoid subjective terminology and use neutral language instead."

Answer:
Removed these terminologies

- Detailed Comment 8: Figures

8.1:
What is the information conveyed by Figure 4? Consider removing it.

Answer:
The Figure is removed.

8.2:
Figure 5.b. : this representation of the wind distribution is unusual. A more standard representation as the probability distribution function would be more meaningful for the reader.

Answer:
The previously numbered Figure 5 is now Figure 4.
Thank you for the feedback. Although the violin plot representation may be less conventional, it was chosen for its ability to convey detailed insights into the distribution of wind power across multiple locations within a single, compact visualization. Overlaid PDF plots were considered; however, they tended to appear cluttered, making it difficult for readers to extract meaningful information. Separate PDF plots for each location were also examined, but they would have required significantly more space. While a CDF plot was another option, its interpretation is less intuitive than the violin plot, especially for readers less familiar with cumulative distributions. To enhance clarity, we have included additional text ( l. 405-418) explaining how to interpret the violin plot, which should aid in understanding its unique presentation.

Changes in paper:
"The wind generation distribution across all locations is available in Fig. 4. This violin plot illustrates the distribution of normalized wind power generation across five different locations (X, A, B, C, and D). Each half-violin represents the density of wind power measurements for a location, showing where values are most concentrated. The symmetrical nature of each violin plot, with mirrored halves for each location, is a standard feature of violin plots that allows for a clearer visualization of the data distribution, where each half represents the same distribution of wind power measurements. The width of each violin indicates the density: wider sections reflect more frequent occurrences of those power levels, while narrower sections suggest less common values. The plot uses a logarithmic scale on the y-axis, making it possible to visualize variations in power generation across a broad range, from very low to high outputs. Inside each violin, the black bar marks the interquartile range, while the white dot represents the median of the wind power measurements for that location. This combination allows for a clear comparison of both the range and central tendencies of wind power output across different sites. For example, a location with a narrower and higher median distribution might experience more consistent and higher wind power generation (i.e., location X), while one with a broader distribution and lower median could have more variability (i.e., location C). Locations A, B, and D share similar distributions where the shape of these

distributions suggests that low power output is more common, with occasional rises to higher values."

8.3:
"Please follow the journal guidelines for the captions: https://www.wind-energy-science.net/submission.html#figurestables "

Answer:
Figures were changed to comply with the guidelines.

8.4:
Consider using intelligible notation in the legend and labels when possible, instead of introducing the notation in the caption. For example, the labels of Figure 7 do not correspond to previously introduced notation.

Answer:
Note that Figure 7 is now Figure 5.
The notations are now introduced in the texts and then used in the figures. The notations are now consistent among all figures, tables, and text.

8.5:
Figure 6,7 and 10: including the equation for the RMSE in the label seems unnecessary since the notation and equations is introduced in the main text.

Answer:
Figures were modified.
Note that Figures 6, 7, and 10 are now A1, 5, and 8.

8.6:
Figure 8: Please indicate the unit on the figure labels.

Answer:
The units are now included.

8.7:
Figure 9: it is unclear why two figures are relevant here. Consider removing Figure 9 (a).

Answer:
Note that Figure 9 is now Figure 7.
Figure 7(a) is shown because it is hard to visualize the extent of the scatter from the PDF plot shown in Figure 7(b). This scatter helps to understand the deviations from the HF EMS, as shown in Figure 6 (previously Figure 8).

8.8:
l. 443: "The mean ($\mu$) being close to zero suggests…": Note that Figure 9(b) indicates that the mean is equal to zero.

Answer:
Text changed to avoid confusion: l. 468.
"The mean ($\mu$) being equal to zero suggests that the surrogate's calculations are unbiased on average"

8.7:
Section 2.1.: a figure illustrating the EMS would be relevant to support its description in the text. For example, Figure 1 could be extended to describe the time schedule for bidding and dispatch decisions.

Answer:
Figure 1 was modified. Additional information was added to the bidding process. Additional text was added to explain further the bidding process of the EMS (l. 204-214). For clarity, throughout the article, the use of the acronym "EMS" was slightly changed: when applicable, "EMS" was modified to "SM optimization", referring to the bidding process happening at the day-ahead stage. The acronym "PMS" was removed entirely and replaced by Real-Time (RT) dispatch. The HF EMS combines both SM optimization and RT dispatch.

Changes in paper:
l. 204-214: "Figure 1 illustrates the considered EMS workflow. The EMS operates through a structured daily cycle, beginning with the forecasting stage. On the previous day (d-1), a forecast of wind generation and spot market prices for the following day (d) is obtained. Using this data, the SM optimization is conducted at noon on day d-1, aligning with the day-ahead market closure, to determine the optimal hourly power bidding for the HPP. This optimization is formulated as a Mixed Integer Linear Programming (MILP) problem, aiming to maximize the plant's revenue across the day by strategically bidding power on the SM.
On day d, the RT dispatch optimization is executed at 15-minute intervals throughout the day. This phase focuses on minimizing discrepancies between the power that was bid on the spot market and the actual real-time available power. The RT dispatch is modeled as a Mixed Integer Quadratic Programming (MIQP) problem, which dynamically adjusts the HPP's output to meet market commitments as closely as possible, responding to variations in generation.
Finally, the day concludes with a settlement process on day d+1, where the outcomes of the day's operations are reconciled. The details of all models can be found in the referenced work."

Figure 1:

[Figure]

**Figure 1.** Workflow of HF EMS of this article, developed by Zhu et al. (2022)

Detailed Comment 9: Equations

9.1:
For the presentation of the equations in the manuscript, consider introducing the relevant metrics and their notations before the equation.
Consider adding a paragraph or a subsection to introduce the notation used in the paper, since there is a wide variety of symbols, subscripts and superscripts in the manuscript.
l. 268-271: please introduce notation in a paragraph and not as a list. This comment applies to subsequent equations as well.

Answer:
Thank you for the feedback. The notations are now introduced in a paragraph before the equations.

9.2:
Equations 1 to 3 are not equations since they don't include an equal sign. Consider giving each scalar parameters a name, a symbol and describe their meaning.

Answer:
Indeed, apologies for the oversight. They are now included in the paragraph instead.

9.3:
The notations "SM" (l.211) and "\lambda" (Eq. 10) are used to describe the price of electricity. Please use a consistent notation throughout the paper.

Answer:
Thank you for pointing it out, the notation $SM_t$ is now used throughout the paper.

9.4:
"Equation 8: consider introducing a specific symbol for the RMSE instead of using the abbreviation."

Answer:
The notation of RMSE was changed to $\varepsilon_{RMS}$ and NRSME to $\varepsilon_{NRMS}$.

Detailed comment 10:

10.1:
Please describe in the abstract that the study was conducted for participation on the day-ahead market and for Denmark.
Please include in the abstract the assumption of perfect forecast.

Answer:
The additional information was added.

10.3:
"Sizing of Hybrid Power Plants (HPPs), which include wind power plants and battery energy systems, is essential to capture tradeoffs among various technology mixes": please be more specific.

Answer:

The tradeoff mentioned is an economic tradeoff, that could lead to over- or under-sizing a HPP.

Changes in paper:
The reformulated first sentence of the abstract.
"Optimal sizing of Hybrid Power Plants (HPPs), which include wind power plants and battery energy systems, is essential to prevent financial losses from under- or over-sizing relative to grid connection capacities"

10.4:
l. 4 "model the operation of a battery when participating in any market": please be more specific about the market mentioned here.

Answer 10.4:
The term was changed to electricity market. Later in the abstract, we mention that the study focuses on the day-ahead market.

10.5:
l. 5: "Traditional EMS" : what does "Traditional" mean here? Consider rephrasing.

Answer:
Based on context, we have Replaced "Traditional EMS" with either High-fidelity EMS or LF EMS.

- Minor comments

All minor comments have been addressed, and changes have been made accordingly.

Concerning the comment:
"Be aware that Wind Energy Science guidelines state that grey-literature may only be cited if there are no alternatives. The international hybrid power plant conference is grey literature, due to its lack of peer-review: "Das, K., Hansen, A. D., Koivisto, M., and Sørensen, P. E.: Enhanced features of wind-based hybrid power plants, Proceedings of the 4th International Hybrid Power Systems Workshop, 2019."
"

We have contacted the Workshop organizers and received the following reply:
"That's partly correct, we only review the abstracts, in the short time between paper deadline and the workshop (about 4-6 weeks) we cannot completely review 180 papers. But those papers which are published in the IEEE Explorer should not be considered gray-literature as IEEE is running them though their quality check... also, in the last few years we published the proceedings in a digital data base, see https://digital-library.theiet.org/content/conferences/cp847, so if you mentioned the ISBN Number in the reference, it should qualify as a reference.

However, we only started with the digital data base in 2021, but all older proceedings also have an ISBN number and the proceedings have been submitted to a number of University library in Europe, so papers could be found by interesting parties. The relevant reference for the 2019 workshop is:

Proceedings 18th International Workshop on Large-Scale Integration of Wind Power into Power Systems as well as on Transmission Networks for Offshore Wind Plants
Dublin, Ireland, 15-16 October 2019
ISBN: 978-3-9820080-5-9"

---

## Author Comment (AC4)

**Reply on RC2 – PDF Format**

- The abstract indicates that the EMS "is introduced to model the operation of a battery," which sounds like a narrower scope than the wind-battery system described in the paper.

Answer:
Changed wording in abstract to expand scope to wind + battery operation.
"Accurate sizing requires high-fidelity Energy Management Systems (EMS) to model bidding strategies and operations in electricity markets…"

- The RMS errors noted in the abstract are not especially meaningful when given as numerical values (to the reader who doesn't yet know what are the scales of the metrics being evaluated). It would be more useful to give these as a percentage or to add the relevant context.

Answer:
Removed RMSE of hourly data and added Normalized RMSE (NRMSE) of yearly revenues in percentages. Added ranges for RMSE of profitability index.

Changes in paper:
l.11- 16: "This surrogate achieves a normalized root mean square error of 0.81% in approximating yearly revenues. This method proves effective in accurately evaluating the operation of HPPs across various geographical locations and hence in multiple sizing problems. Furthermore, we utilize the surrogate to evaluate the profitability of several HPP sizes, achieving a root mean square error of 0.010 on the profitability index, with values ranging between -0.13 and 0.18. This demonstrates that the developed surrogate model is suitable for HPP sizing under the given cost and financial assumptions."

- Table 1 certainly motivates that using a high-fidelity EMS can be computationally expensive, but without direct comparisons in your own results it is difficult to assess the relative value of the SM approach.

Answer:
We have now removed this Table and replaced it with text in l. 78-100 where the computational burden of the high-fidelity EMS is detailed. The comparison between the HF EMS and the surrogate is highlighted in Section 5.4 and Section 6 l. 541-546.

Changes in paper:

l-78-100: "The trade-off between computational efficiency and model accuracy presents a significant challenge for the optimal sizing of HPPs. A sizing optimization of an HPP involves maximizing a financial metric by varying the wind power plant rating, battery energy, and power ratings. Calculating that financial metric requires solving an EMS model for each potential HPP configuration. Consequently, HF EMS models offer precise assessments; however, relying solely on them is impractical due to their substantial computational demands. Conversely, using LF EMS models reduces computational time but risks compromising the financial viability of the project due to inaccurate assessments.
To illustrate the computational burden of an HF EMS model, we evaluate the state-of-the-art EMS developed by Zhu et al. (2022) This model requires 1,250 minutes to solve for 25 years of operation (the assumed lifetime) of a given HPP using a single-node High Performance Computing (HPC)

cluster, Sophia (DTU HPC Cluster, 2019), which has 32 physical cores (2 × sixteen-core AMD EPYC 7351) and 128 GB of RAM (4 GB per core, DDR4@2666 MHz). Therefore, even if we need to evaluate only a few sizings for the optimizer to converge, we require a substantial amount of time to reach a solution. For example, evaluating 10 sizings takes 12,500 minutes, or approximately 208 hours. Additionally, in previous work familiar to the authors Leon et al. 2024, a sizing optimization can take up to several hundred iterations to approach optimality. In that study, the authors use a low-fidelity EMS model to evaluate the operation of an HPP over its lifetime in a matter of 15 seconds. The comparison of the optimization time is based on the same computational resources.

Given these computational benefits, HPP sizing optimization often relies on LF EMS models. For example, Leon et al. 2024 propose a methodology for sizing HPPs as a nested optimization problem, using two LF EMS models: a short-term EMS formulated as linear programming and a long-term rule-based EMS. The short-term EMS provides a baseline for daily optimal operations, while the long-term EMS modifies these operations to account for degradation effects and forecast inaccuracies over the plant's lifetime. Similarly, in a study aimed at optimizing the design and layout of a hybrid wind-solar-storage plant, Stanley and King (2022) employs a simple battery dispatch model, where the battery is only discharged to meet minimum power requirements. While using LF EMS models may result in reduced accuracy in revenue estimation, they are widely adopted in HPP sizing due to computational efficiency. Indeed, several review studies underscore the prevalence of LF EMS models in sizing methodologies (Roy et al., 2022; Lian et al., 2019; Thirunavukkarasu et al., 2023)"

l-541-546: "The fast and accurate surrogate allows us to evaluate an HPP's profitability throughout its lifetime with little computational burden. Indeed, the surrogate model is capable of evaluating the NPV/CAPEX for all 50 HPP configurations in Fig. 10 in 25 seconds. In contrast, computing the same evaluations using the high-fidelity model for each HPP configuration, with inputs spanning over a year, would take approximately 39 hours. However, it is important to understand the impact of the surrogate's accuracy on the PI. Figure 10 shows that the surrogate can be reliably used if slight deviations of around 0.010 in the PI are acceptable for the intended business evaluation. In other words, the error on the computed NPV is around 1% of the CAPEX."

- Many abbreviations are defined multiple times (e.g., SM on both lines 113 and 122); please check that all are defined only the first time they are used. (I see PI and HPP re-defined as late as p. 18.)

  Answer:
  Changes were carried across the paper.

- Dispatch intervals are given as both 15 min (line 126) and 5 min (line 135). I assume from other parts that 5 min is a typo but please clarify if not.

  Answer:
  Indeed, it was a typo. Thank you for pointing it out.

- There are several 1-sentence paragraphs that interrupt the flow. For example the sentence on line 139 (introducing Figure 1) could easily be combined with the paragraph starting on line 140 (and similar in subsequent instanced).

  Answer:
  All 1-sentence paragraphs are now combined with their corresponding paragraphs.

- Table 3 clearly indicates many variables and constraints but it would be useful to tie these more explicitly to the computational burden noted as a goal for the proposed surrogate model. Is this table related to the 47-min computation noted on line 148?

  Answer:
  These indeed refer to the 47-minute computation time. In the text, we explain that each iteration of the MILP and MIQP problem is solved quickly, in less than 0.15 seconds (l. 220). However, many of these optimization problems must be solved sequentially, leading to the 47-minute computation time for one year of input data. The text was slightly modified to make it more explicit that we are referring to the HF EMS on which the surrogate is based.

  Changes in paper:
  l. 216-224:
  "It was observed that for a given HPP configuration, 47 minutes were required to compute the outputs for one year of operation of the HF EMS model. The underlying reason for this is due to the iterative and sequential nature of the framework. For each day, the MILP optimization is solved first, followed by the MIQP for each dispatch interval (e.g., 96 times per day). While each iteration of the MILP and MIQP problems requires a minimal amount of time (less than 0.15 seconds), the frequency of these optimizations is substantial. Moreover, since each time step depends on the previous one, it is necessary to perform the optimization sequentially. Table 3 shows the number of decision variables and constraints required to optimize for inputs spanning over one year. This highlights the substantial computational time required to optimize the sizing of an HPP based on such an operational model."

- (1)-(3) are ratios, not equations as stated. Either give them (short) variable names or omit the equation treatment (you use the ratios as is later in the paper, e.g., line 205)

  Answer:
  Thank you for noting it. The ratios are included in the text, and the equations were removed.

- I struggled at times to understand which surrogates were being described and analyzed. It would be very helpful to give the four surrogates in Table 5 names (e.g., S1, S2, etc.) and then use these names consistently throughout the rest of the paper (e.g., "…surrogate S1…")

  Answer:
  Thank you for the suggestion. The surrogates have been named S1-S4, and the text, figures, and tables were modified accordingly.

- Captions in general are quite short and could be more descriptive. As one example, it would be easier to understand Figure 2 if the 2 sentences on lines 210-211 explaining the nomenclature were in the caption instead of the body text.

  Answer:
  The captions have been changed so that most figures are more descriptive. Additionally, all metrics and variables are now explained before each figure.

- Also in Table 5 I assume that "FFN" is a typo and it should be "FNN"; otherwise, please explain.

  Answer:
  Indeed, it is a typo.

- It is not clear what would be the desired level of truncation for the principle component matrices Z (line 220); how was this desired level selected?

  Answer:
  Thank you for pointing that out, I apologize for the oversight. Additional text is now added to explain that the truncation level is such that we have an explained variance of 99%: l. 274-278.

Changes in paper:
l. 274-278: "After applying Singular Value Decomposition (SVD) to both matrices, $M_{in}$ and $M_{out}$, we extract their principal component matrices and truncate them to the desired level. As a result, we obtain two sets of matrices with different truncation levels, – denoted as $r_{in}$ and $r_{out}$, respectively. The truncation level is chosen so that the explained variance is 99%; for the definition, see Eq. 4 of Freire and Ulrych (1988). Table 6 presents an overview of the features and samples of each data-processing method for input and output data spanning over a year."

- Line 257 is missing the word "Appendix"

  Answer:
  Word added.

- Please clarify if the y terms in (8) are for the normalized or actual values in the time series. Line 266 suggests normalized but 269 and 270 discuss true and predicted data without the normalization qualifier. This will also impact the quality of the results as measured by RMSE (i.e., relative to a scale of 0-1 or a much wider scale from the original data).

  Answer:
  Note that Eq. (8) is now Eq. (5).
  The y-terms refer to the normalized values. The variables within the RMSE' equation (5) are now modified for clarity. Additionally, the figures now show explicitly when the normalized variables are used.

- Why is the text below (9) only appearing in a subset of the page width?

  Answer:
  It is now integrated into a paragraph.

- Line 300 explains PI in words, but the equation doesn't appear until approximately a half page later; could be more streamlined to just have the equation in the paragraph where it is introduced.

  Answer:
  Thank you for the suggestion. We have included the equation for the PI in the paragraph.

  Changes in Paper:
  l. 338-337: "A more meaningful measure is the Profitability Index (PI), calculated as *NPV/CAPEX*."

- I recommend avoiding use of longer, non-standard words as "symbols" in equations, e.g., "Profit_y" in CF_y on line 319, as this makes the equations harder to read. (I understand that CAPEX and OPEX are often used as such in equations in wind energy related publications.)

  Answer:
  We have changed some variables according to the comment.

- Line 341: I assume it should be $(y\_b(i\_b))$ (subscripts) as in the equation.

  Answer:
  Indeed, thank you for pointing it out.

- Lines 354-355: these variables have been previously defined

  Answer:
  This is now corrected.

- Lines 362-363: I don't know what "This tool is based on re-analyzes…" means. Re-analyzed?

  Answer:
  There was a typo; it is now changed to "meteorological reanalysis data."

- Please ensure adequate font size in all figures (especially Figure 4)

  Answer: this has been modified.

- Typo in Figure 4b caption (should be Normalized prices)

  Answer: Figure 4 is now removed.

- Section 4.1.3: are 250 or 200 HPP configurations studied? The end of p. 16 says both.

  Answer:
  The paragraph of this section is now modified for clarity: there are a total of 250 HPP configurations used, 200 for training and 50 for validation. All configurations are unique.

  Changes in paper:
  l.376-378: "To ensure an equal distribution of all variables across the entire parameter space, the Latin Hypercube Sampling method, by Jin et al. (2005), is used to randomly select 250 sizing configurations, of which 200 HPP (80%) are used to train the regressor and 50 HPP (20%) are used to evaluate the accuracy of the surrogate, as detailed in Section 2.2.2."

- Figure 5(b) is an interesting way to visualize the different probability distributions but needs explanation since it is non-standard. Also, the y-axis needs units (MW?)

Answer:
An additional explanation is now included to explain the plot: l. 413-426. As the y-axis has normalized wind power generation, it is unitless.

Changes in paper:
l.415-418: "The wind generation distribution across all locations is available in Fig. 4. This violin plot illustrates the distribution of normalized wind power generation across five different locations (X, A, B, C, and D). Each half-violin represents the density of wind power measurements for a location, showing where values are most concentrated. The symmetrical nature of each violin plot, with mirrored halves for each location, is a standard feature of violin plots that allows for a clearer visualization of the data distribution, where each half represents the same distribution of wind power measurements. The width of each violin indicates the density: wider sections reflect more frequent occurrences of those power levels, while narrower sections suggest less common values. The plot uses a logarithmic scale on the y-axis, making it possible to visualize variations in power generation across a broad range, from very low to high outputs. Inside each violin, the black bar marks the interquartile range, while the white dot represents the median of the wind power measurements for that location. This combination allows for a clear comparison of both the range and central tendencies of wind power output across different sites. For example, a location with a narrower and higher median distribution might experience more consistent and higher wind power generation (i.e., location X), while one with a broader distribution and lower median could have more variability (i.e., location C). Locations A, B, and D share similar distributions where the shape of these distributions suggests that low power output is more common, with occasional rises to higher values."

- Please ensure that all results in Section 5 (figures, tables, and text) are clear about which surrogate model has been used (referring to the suggestion to give them names in Table 5)

Answer:
The results in this section now refer to the best-performing surrogate, model S4. This has been clarified in the text. The comparison of the performance of all surrogate models is now moved to Appendix A.

- For Figure 6 and 7 (and related discussion), I refer back to my question about whether the RMSE is based on the normalized data to help the reader evaluate the quality of the method.

Answer:
Note that Figures 6 and 7 are now Figures A1 and 5.
For both figures we use normalized data. For Figure 5, we now explicitly mention it in the text and on the figure by using the normalized variables as the x-axis labels. For figure A1 we mention in the text leading up to the figure that we use the normalized data.

- For Figure 8, instead of noting "MegaWatts" in the caption it would be better to include "(MW)" in each of the y-axis labels

Answer:
This has changed.

- On line 435 and related discussion you mention the "density" of the data points but the colorbar on Figure 9(a) has units of "count". I understand that these are related but more precise language would be more clear.

  Answer:
  This has been clarified in the text: l. 458-459.

  Changes in paper
  "The hexagonal bins group nearby points (denoted as count in Fig. 7(a)) and show the density of data points within each bin."

- Line 477: please name the surrogate used instead of "the selected surrogate" here, as well

  Answer:
  This has been modified as suggested.

- Line 515: "…all HPP configurations are not profitable…" has a different meaning than "…not all HPP configurations are profitable." I think you mean the latter and should therefore revise accordingly.

  Answer:
  This has been modified according to the suggestion. Thank you.

- Typos and grammar to change:
  -line 17: "wind power plants are" (should be plural)
  -line 20 appears to be missing a space between ".This"
  -line 42: "accurate forecasting can mitigate these penalties" should be proceeded by ; (not a comma) or a standalone sentence
  -Table 1: "Iterations" should be plural
  -line 90: the sentence starting "Two of which" is incomplete
  …and so on. I recommend a close re-reading as part of the revision process to address these and similar errors throughout.
  Answer:
  Several of these mistakes have been modified after a closer re-reading.

- Furthermore, I believe the citation format is not aligned with WES standards (Author, Year) in most cases except where the author's name is part of the sentence (e.g., "Author (year) showed that…").

  Answer:
  Several re-readings and editing have been carried out to correct for theses mistakes.